



# The Effect of Construction Material on the Thermal Gain Dependence of a Fluxgate Magnetometer Sensor

David M. Miles[1], Ian R. Mann[1], Andy Kale[1], David K. Milling[1], Barry B. Narod[2,3], John R. Bennest[4], David Barona[1], Martyn J. Unsworth[1]

[1]University of Alberta, Edmonton, AB, Canada
[2]Narod Geophysics Ltd., Vancouver, BC, Canada
[3]University of British Columbia, Vancouver, BC, Canada
[4]Bennest Enterprises Ltd., Summerland, BC, Canada

*Correspondence to*: David M. Miles (david.miles@ualberta.ca)

**Abstract.** Fluxgate magnetometers are an important tool in geophysics and space physics but are typically sensitive to variations in sensor temperature. Changes in instrumental gain with temperature, thermal gain dependence, are thought to be predominantly due to changes in the geometry of the wire coils that sense the magnetic field. Scientific fluxgate magnetometers typically employ some form of temperature compensation, and support and constrain wire sense coils with bobbins constructed from materials such as MACOR machinable ceramic (© Corning) which are selected for their ultra-low thermal deformation rather than for robustness, cost, or ease of manufacturing. We present laboratory results comparing the performance of six geometrically and electrically matched fluxgate sensors in which the material used to support the windings and for the base of the sensor is varied. We use a novel, low-cost thermal calibration procedure based on a controlled sinusoidal magnetic source and quantitative spectral analysis to measure the thermal gain dependence of fluxgate magnetometer sensors at the part-per-million per degree Celsius level in a typical magnetically noisy university laboratory environment. We compare the thermal gain dependence of sensors built from MACOR, polyetheretherketone (PEEK) engineering plastic (virgin, thirty percent glass filled, and thirty percent carbon filled), and Acetal to examine the trade between the thermal properties of the material, the impact on the thermal gain dependence of the fluxgate, and the cost and ease of manufacture. We find that thermal gain dependence of the sensor varies as one half of the material properties of the bobbin supporting the wire sense coils rather than being directly related as has been historically thought. An experimental sensor constructed from thirty percent glass filled PEEK (21.6 part-per-million per degree Celsius) had a thermal gain dependence within 5 part-per-million per degree Celsius of a traditional sensor constructed from MACOR ceramic (8.1 part-per-million per degree Celsius). If a modest increase in thermal dependence can be tolerated or compensated, then thirty percent glass filled PEEK is a good candidate for future fluxgate sensors as it is more economical, easier to machine, lighter, and more robust than MACOR.





## 1 Introduction

Fluxgate magnetometers (Primdahl, 1970) are widely used in geophysics and space physics to measure static and low-frequency magnetic fields. However, they have long been known to be sensitive to the temperature of the sensor (Trigg et al., 1971) with the dominant effect thought to be a change in gain with temperature due to variations in the geometry of the coils

of wire used to sense the magnetic field. In particular, fluxgate sensors measure the static magnetic field by periodically driving a ferromagnetic element (core) into magnetic saturation, and then detecting the resulting change in the magnetic field as current or voltage in a surrounding sense winding. Negative magnetic feedback can be provided by driving current back into either the sense winding itself or into a separate feedback winding. Fluxgate magnetometers can be affected by temperature in a variety of ways, including: alteration of the magnetic properties of the core, mechanical stress on the core due to thermal

mismatch between the ferromagnetic core and its support structures, change in the geometry of the ferromagnetic core, change in the geometry of the sense windings, changes in the geometry of the feedback windings, changes in the orthogonality or alignment of the sense windings, changes in the resistance of the sense or feedback windings, or changes in the drive current used to saturate the ferromagnetic core. However, the dominant factor has historically been taken to be the thermal expansion of the sense and feedback windings (Acuña et al., 1978). Specifically, expansion or contraction of the bobbin with temperature

changes the winding density (turns of wire per unit length), modulating the sensitivity of the coil. Expansion or contraction also causes changes to the cross-section area of the sense coil which may introduce another temperature effect in low aspect ratio windings such as in miniaturised sensors (Miles et al., 2016).

Primdahl (1970) and then Acuña et al. (1978) described a method whereby the temperature dependent resistance of the feedback winding was successfully used to compensate for temperature dependent variation in the feedback coil dimensions.

Other geometries and coil topologies have been explored with the intention of minimising cross-axis effects by creating a 'magnetic vacuum' within the sensor where the field is homogeneous and zeroed in all components (Primdahl and Jensen, 1982). However, almost all designs rely on materials with ultra-low coefficients of linear thermal expansion such as quartz or MACOR machinable glass ceramic (© Corning), to minimise the thermal effects and to allow linear temperature compensation to be successful.

MACOR machinable ceramic has been used extensively and successfully in a variety of fluxgate applications. The specific materials used in sensor construction are often not provided in instrument publications. However, MACOR is explicitly mentioned or known to be used in: the NASA MAGSAT satellite (Acuña et al., 1978); the S100, STE and PC104 observatory magnetometers developed by Narod Geophysics Ltd. (Narod and Bennest, 1990) and used in both the Canadian CARISMA ground network (Mann et al., 2008) and the US EMScope magnetotelluric network (Schultz, 2009); the Danish Oersted satellite

(Nielsen et al., 1995); the miniaturised SMILE instrument (Forslund et al., 2007); a prototype radiation tolerant fluxgate (Miles et al., 2013); and the Canadian Space Agency Cassiope/e-POP satellite (Wallis et al., 2015). Unfortunately, MACOR is expensive, difficult to machine, and brittle. Several authors have recently begun to use modern polyetheretherketone (PEEK)


engineering plastic, either virgin or partially filled with glass or carbon, for fluxgate sensors (Butvin et al., 2012; Miles et al., 2016; Petrucha et al., 2015; Petrucha and Kašpar, 2012). PEEK derivatives should be a less expensive, easier to manufacture, and more robust alternative, albeit with a larger thermal expansion coefficient. The authors wanted to isolate and measure the effect on thermal gain dependence of changing different components of the sensor from MACOR to PEEK based plastic

alternatives. This paper presents laboratory results comparing the performance of six geometrically and electrically matched fluxgate sensors in which the material used to support the windings and for the base of the sensor is varied. The goal is to construct a sensor which is more robust, has a lower materials cost, and is easier and less expensive to manufacture without significantly compromising the thermal gain stability of the instrument.

## 2 Fluxgate Theory

### 2.1 Introduction

A fluxgate magnetometer (Primdahl, 1970) assembles a vector magnetic measurement from three solenoidal sense windings, each sampling an independent orthogonal component of the magnetic field. Each solenoid contains a ferromagnetic core which concentrates the ambient magnetic field, and which is periodically forced into magnetic saturation to modulate (gate) the field experienced in the sensor. This gating creates time varying magnetic flux, which in turn induces electromotive force in the

solenoidal sense winding. Figure 1 illustrates one axis of a ring-core implementation: the ferromagnetic core is formed into a closed ring, the gated field creates electromotive force in a rectangular solenoidal sense winding, and a toroidal drive winding minimises the transformer coupling between the saturating current in the drive winding and the resulting fluxgate signal output by the sense winding.





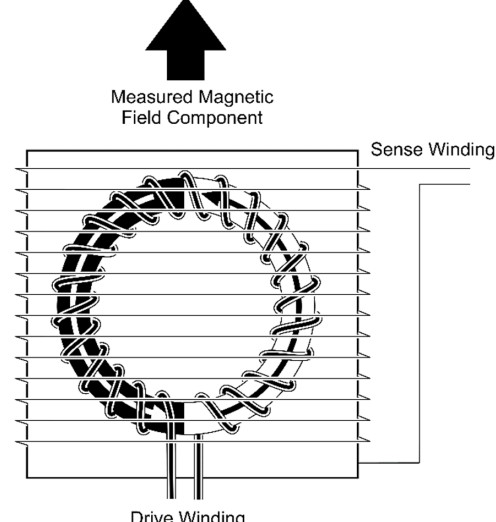

Figure 1: Schematic illustration of a single sensor axis of a ring-core fluxgate magnetometer.

## 2.2 Expected Effect of Temperature-Induced Changes in Geometry

Global negative feedback can be used to linearize a fluxgate magnetometer, and to increase the range of magnetic fields that

5    can be sensed without saturating the instrument. A feedback current, $I_F$, proportional to the measured magnetic field on each axis, is driven back into the sense winding to force the average magnetic field along that axis towards zero. Changes in the geometry of the sense windings will therefore create a thermal dependency in two ways: by affecting the forward gain of the solenoid as a sensor; and, more significantly, by affecting the feedback gain which scales the conversion of feedback current, $I_F$, into feedback flux, $B_F$. Following and expanding on the approach in Acuña et al. (1978), we approximate the sense winding

10    as a finite solenoid (Figure 2) of n turns, length L, and radius R.





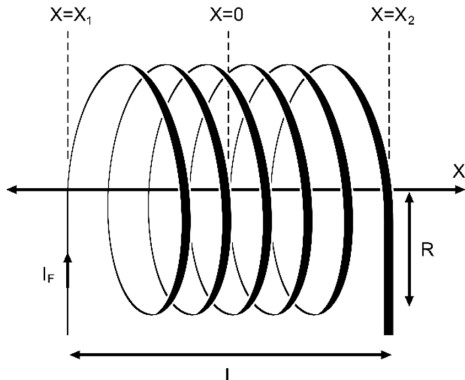

Figure 2: Approximating the sense winding as a finite solenoid of length L and radius R.

In this expanded Acuña et al. (1978) approximation, we consider the magnetic field along the axis of the solenoid, x. The field within the solenoid, $B_F$, is then dependent on the permeability of free space, $\mu_o$, the current in the coil, $I_F$, and is given by

$$B_F = \frac{\mu_o n I_F}{2L} \left( \frac{x - x_1}{\sqrt{(x - x_1)^2 + R^2}} - \frac{x - x_2}{\sqrt{(x - x_2)^2 + R^2}} \right)$$

(1)

The sensor output will effectively be the volume integral of the magnetic flux inside the sense winding. This approximation ignores the complicating factor that flux is concentrated by the ferromagnetic ring as it periodically enters and leaves magnetic saturation. For simplicity, and given the limitations of this approximation, the overall trend of the sensor is assumed to match that of a point at the center of the coil on the solenoidal axis ($x_2 = -x_1$, $x = 0$). The field is then

$$B_F = \frac{\mu_o n I_F}{2L} \left( \frac{-x_1}{\sqrt{(-x_1)^2 + R^2}} - \frac{x_1}{\sqrt{(-x_1)^2 + R^2}} \right) = \frac{-\mu_o x_1 n}{\sqrt{(-x_1)^2 + R^2}} \frac{I_F}{L}$$

(2)

For simplicity, we define K to collect variables and note that $x_1 = R$ at the center of a square winding such as is shown in
Figure 1. Therefore,

$$K = \frac{-\mu_o x_1}{\sqrt{(-x_1)^2 + R^2}} = \frac{-\mu_o x_1}{\sqrt{x_1{}^2 + x_1{}^2}} = \frac{-\mu_o}{\sqrt{2}}$$

(3)

which is constant with temperature. Note that the ferromagnetic ring-core shown in Figure 1 is bonded to a supporting metal ring that also has a temperature dependence. This introduces additional potential temperature dependencies in that the geometry of the ring may change, or the ring may deform the geometry of the sense winding – both are ignored in this approximation. The effect of temperature, T, can be included by assuming that the length, L, of the coil is controlled by the bobbin on which
it is wound. L will therefore vary around an assumed length, l = L when T = 0, due to the coefficient of linear thermal expansion of the bobbin material, $\alpha_m$.


$$L = l(1 + \alpha_m T) \qquad (4)$$

Substituting Eq. (3) and Eq. (4) into Eq. (2) gives

$$B_F = \frac{K I_F}{l(1 + \alpha_m T)} \qquad (5)$$

which recreates the result from Acuña et al. (1978) . To quantify the effect of temperature on instrument sensitivity, we define $\alpha_g$, the coefficient of thermal gain dependence (ppm °C$^{-1}$), as the change in the measured amplitude of a fixed test signal with sensor temperature. This coefficient is then related to the sensor materials via their coefficients of linear thermal expansion, $\alpha_m$

(ppm °C$^{-1}$), which is a manufacturer provided estimate of the expansion or contraction of the material with temperature. Mathematically, $\alpha_g$ can be expressed as

$$\alpha_g = \frac{1}{B_F} \frac{dB_F}{dT} = \frac{l(1 + \alpha_m T)}{K I_F} \frac{-K \alpha_m I_F}{l(1 + \alpha_m T)^2} = \frac{\alpha_m}{\alpha_m T + 1} \qquad (6)$$

$\alpha_m$ ranges from $8.1 \times 10^{-6}$ to $85 \times 10^{-6}$ for each of MACOR, the PEEK derivatives, and Acetal (see Table 1 and associated text for material datasheet references). Therefore, $\alpha_m T \ll 1$, and to a good approximation

$$\alpha_g = \frac{\alpha_m}{\alpha_m T + 1} \approx \alpha_m \qquad (7)$$

This was first proposed by Acuña et al. (1978), and indicates that, to a first approximation, the coefficient of thermal gain

dependence of the sensor should be equal to the coefficient of linear thermal expansion of the bobbin on which the sense winding is wound. These approximations are linked to the geometry of the sensor, and do not easily generalise. Other approximations are possible, including modelling the sense windings as a flat solenoid.

### 2.3 Electronic Temperature Compensation

Acuña et al. (1978) described a method of temperature compensating the feedback current, $I_F$, to correct the linear component

of the sensor's temperature dependence. Figure 3 shows simplified feedback amplifier design, adapted from Acuña et al. (1978) and used in the instrument described herein where a transconductance amplifier has been modified such that the normally constant voltage-to-current transfer function is engineered to respond to the load resistance. The appropriate scaling of this transfer function allows, to first order, the effect of temperature on the load resistance offered by the sense winding to compensate for the effect of temperature on the sensor geometry.


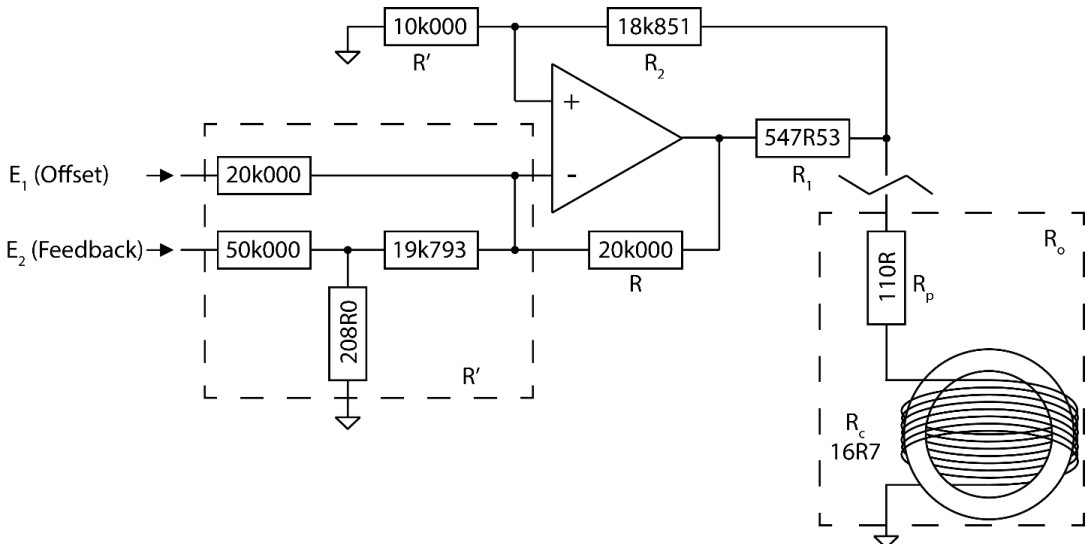

Figure 3: Modified transconductance amplifier providing temperature compensation to a single magnetometer channel.

The Acuña et al. (1978) design used platinum wire for the windings, whereas here the windings are copper in series with a platinum resistor. Detailed analysis of this circuit design was first prepared in an informal technical report (Narod, 1982) which

5  has been reproduced and expanded in Appendix A. Based on Eq. (A10) in Appendix A, and expanding all terms using the nomenclature in Figure 3, the temperature compensation of the feedback amplifier is given by

$$\frac{1}{I_F}\frac{dI_F}{dT} = \frac{-\alpha_p \left[\frac{R_1 + R_2 - R}{R' + R_2}\right] R_o}{R_1 + \left[\frac{R_1 + R_2 - R}{R' + R_2}\right] R_o} \qquad (8)$$

The coefficient of thermal resistivity of the sense/feedback winding and series platinum resistor is well approximated by ($\alpha_p = 3.93 \times 10^{-3}$ °C$^{-1}$).

## 3 Method

10  ### 3.1 Experimental Fluxgate Sensors: Testing Different Structural Materials

Six geometrically and electrically matched fluxgate sensors were constructed where the material used to support the windings and for the base of the sensor was varied. Virgin PEEK, 30% glass filled PEEK, or 30% carbon filled PEEK are candidate materials to replace MACOR in new sensors due to their temperature stability, robustness, ease of machining, and cost. Acetal is included as a control with a large coefficient of linear thermal expansion. Table 1 summarises the properties of the materials

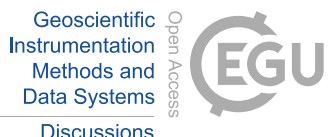

used in this experiment. The MACOR was all manufactured by Corning© and the manufacturer's datasheet is assumed to apply to all samples. Three PEEK materials were procured from Professional Plastics (http://www.professionalplastics.com): virgin PEEK (SPEEKNA2.000D), 30% glass fiber PEEK (SPEEKGL30NA.500), and 30% carbon fiber PEEK (SPEEKCF30.375). The acetal (Delrin) sensor used existed from an earlier experiment so the exact plastic used in its

construction is unknown and hence no manufacturer's datasheet was available. References values for general purpose acetal were assumed (Oberg, 2012). Properties for the Inconel x750 used to support the ferromagnetic core were taken from several sources including the Special Metals Group of Companies datasheet (© Special Metals Group of Companies, Unified Numbering System for Metals and Alloys reference UNS N07750).

Table 1: Properties of the materials used in the sensors.

| Material | Coefficient of Linear Thermal Expansion (ppm °C$^{-1}$) | Young's Modulus (GPa) | Density (g/cm$^3$) |
|---|---|---|---|
| MACOR | 8.1 | 66.9 | 2.52 |
| 30% Carbon PEEK | 18.0 | 9.7 | 1.41 |
| 30% Glass PEEK | 21.6 | 6.9 | 1.51 |
| Virgin PEEK | 46.8 | 3.5 | 1.31 |
| acetal | 85.0 | 3.0 | 1.41 |
| Inconel x750 | 12.6 | 214 | 8.28 |

All the sensors in this experiment use the physical dimensions, design, and construction of the Narod Geophysics Ltd. STE magnetometer (Narod and Bennest, 1990), with three ring-core sensors mounted orthogonally on a common base (Figure 4). This matches the geometry used by Acuña et al. (1978).

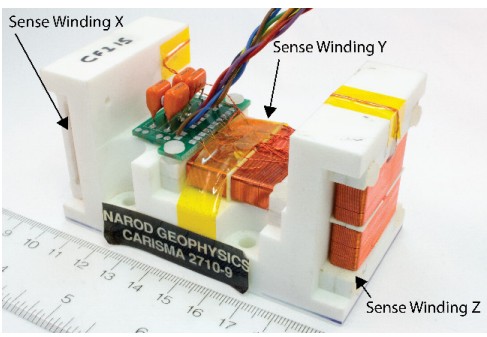

Figure 4: The Narod Geophysics Ltd. STE observatory magnetometer constructs a measurement of the vector magnetic field
from three orthogonal sense windings.

The magnetic flux experienced by each of the three sense windings is varied by periodically saturating a 25.4 mm ferromagnetic ring-core composed of Permalloy foil wrapped on an Inconel x750 support. The ring-core is covered with a single layer of Kapton, and then wrapped toroidally with approximately 350 turns of Belden© 8056 AWG 32 magnet wire to form the drive winding used to saturate the ring-core. The toroidally wound ring-core is positioned on a centering disk within



a rectangular bobbin. The outside of the bobbin supports a separate solenoidal winding, made from 360 turns of Belden© 8056 AWG 32 wire in two sections (Figure 5), which serve as both the sense and feedback coils. Figure 5 shows the length, L, defined by the two channels in the bobbin that contain the parallel sense windings. This length is analogous to that shown schematically in Figure 2.

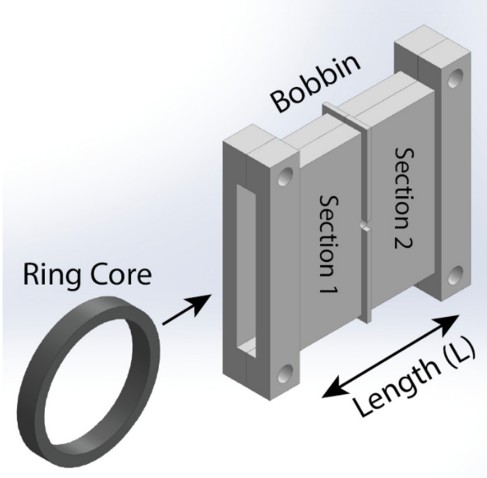

Figure 5: Sensor axis constructed from a ferromagnetic ring core with a toroidal drive winding inserted in a bobbin which supports the combined sense/feedback windings.

Figure 6 shows the six sensors used in this experiment. The name, composition, and roll in the experiment are shown in Table 2.

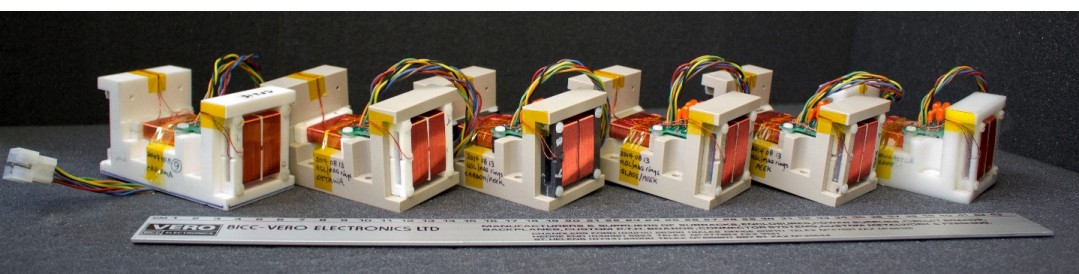

Figure 6: Fluxgate sensors used in this experiment: (Left to Right) MACOR Bobbin / MACOR Base, MACOR Bobbin / PEEK Base, Carbon PEEK Bobbin / PEEK Base, Glass PEEK Bobbin / PEEK Base, PEEK Bobbin, PEEK Base, acetal Bobbin / acetal Base. Note the various colours of material used in the winding bobbins and the base. For example, third from left is Carbon PEEK Bobbin / PEEK Base, distinguished by charcoal-coloured bobbins. The 914.4 mm long ruler is included for
15  scale.



MACOR is considered the reference material for this experiment. A standard STE fluxgate sensor constructed from a MACOR bobbin and a MACOR base (referred to as MACOR/MACOR) was therefore used as the reference against which to compare the other sensors. A MACOR sensor on a virgin PEEK base (MACOR/PEEK) was constructed to distinguish between the effect of the change in dimensions of the sense winding and the change in orthogonality of the three sense windings due to

5    deformation of the base. Three sensors were constructed with virgin PEEK bases and winding bobbins constructed from each PEEK type: virgin PEEK (PEEK/PEEK), 30% carbon filled PEEK (Carbon/PEEK), and 30% glass filled PEEK (Glass/PEEK). A pre-existing sensor with an acetal base and acetal sense windings (acetal/acetal) was used as a negative control due to its known poor temperature stability (large coefficient of linear thermal expansion).

Table 2: Sensors used in this study and their makeup.

| Sensor Name | Bobbin Material | Base Material | Role in Experiment |
|---|---|---|---|
| MACOR/MACOR | MACOR | MACOR | Reference Instrument |
| MACOR/PEEK | MACOR | PEEK | Discriminate between bobbin and base effects |
| Carbon/PEEK | 30% Carbon PEEK | PEEK | Quantify performance of bobbin material. |
| Glass/PEEK | 30% Glass PEEK | PEEK | Quantify performance of bobbin material. |
| PEEK/PEEK | PEEK | PEEK | Quantify performance of bobbin material. |
| Acetal/Acetal | acetal | acetal | Control with poor thermal stability. |

### 3.2 Fluxgate Electronics

All the sensors in this experiment were driven and sampled by a single, unmodified set of STE magnetometer electronics. The STE magnetometer uses a classic second harmonic analog fluxgate design (Geyger, 1962), with its range expanded by the addition of variable offset feedback current (Figure 7). The ferromagnetic ring core is driven at a fundamental frequency $f =$

15    15.625 kHz, creating a fluxgate signal at $2f = 31.250$ kHz. The voltage from the pickup windings is capacitively coupled to block any quasistatic feedback current, and is bandpass filtered twice at $2f$ using tuned passive resistor, inductor, and capacitor (RLC) filters. A phase locked analog switch inverts every other half wave period, creating a synchronous detector to demodulate the sensor output. Finally, a low pass filter and analog integrator create an analog voltage proportional to the magnetic field in the sensor, which is then captured by the analog to digital converter (ADC).





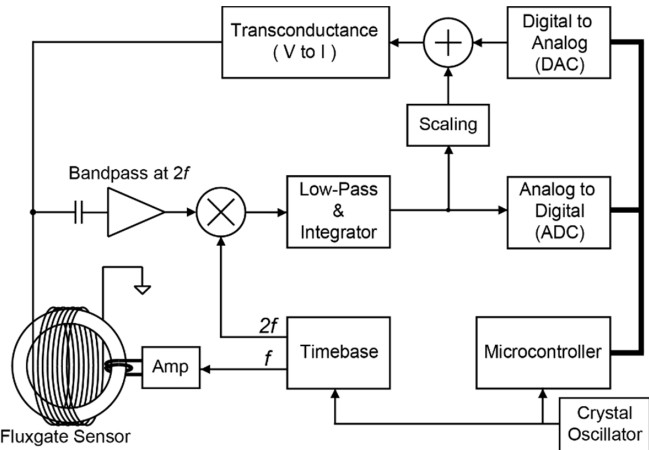

Figure 7: Single channel block diagram of the fluxgate magnetometer. Adapted from Wallis et al. (2015).

This output voltage is attenuated using a resistor T-network (Figure 3), and is summed with an offset voltage from the digital to analog converter (DAC) to create an analog feedback voltage. A modified transconductance amplifier (Appendix A)

converts this voltage into a temperature compensated feedback current that is driven back into the pickup windings, nulling the magnetic field in the sensor. The global negative feedback loop, made up of the analog output of the sensor and the applied offset, is thereby temperature compensated by the transconductance amplifier. The measured value of the magnetic field is then the scaled sum of the offset applied with the DAC and the residual field digitised by the ADC.

The feedback amplifier in the STE magnetometer should give a temperature correction of 19.1 ppm °C$^{-1}$ using Eq. (8) and the

resistor values from Figure 3. This is approximately twice the correction suggested by the Acuña et al. (1978) calculation for the standard MACOR/MACOR sensor ($\alpha_g \approx \alpha_m = 8.1$ °C$^{-1}$). This correction was determined empirically from temperature testing completed in the 1980s. The experiment completed here examines the validity of both the temperature compensation applied by the modified transconductance amplifier and the Acuña et al. (1978) approximation of sensor's temperature dependence.

**3.3 Experimental Setup**

Accurately measuring the effect of temperature on fluxgate performance is technically challenging. The sensor assemblies have both sufficiently high thermal mass and low thermal conductance that the cooling and heating cycles must be slow (hours) to ensure that the sensor temperature is homogeneous, and that the temperature probe is at the same temperature as the sensor itself. On these timescales, the natural variations in the Earth's magnetic field are large compared to the thermal effects being

characterized. The sensor can be isolated from the Earth's varying field using expensive purpose-built nested high-permeability shields. However, temperature variations can change the dimensions of the shields, causing variation in the leakage field that



penetrates the shield. Temperature characterisation is usually completed by either: placing a calibration coil inside a thermally regulated chamber within a magnetic shield, and ensuring that the fixtures are thermally isolated (e.g., the temperature test facility at the Magnetometer Laboratory at the Institute for Space Research in Graz, Austria); or in a magnetically quiet location with active compensation for Earth field variation (e.g., National Resources Canada {NRCan} Geomagnetic Laboratory

Building 8 in Ottawa, Canada).

Here, we demonstrate a novel and low-cost method of measuring thermal gain sensitivity at the ppm °C$^{-1}$ level in an uncontrolled, magnetically noisy laboratory using a simple controlled sinusoidal source and apply the technique to characterising and comparing sensor constructed from different materials. As fluxgate sensor under test changes temperature its gain is affected causing the measured amplitude of the constant test signal to vary. Quantitative narrow frequency band

spectral analysis is used to isolate and measure the apparent amplitude of the test signal, irrespective of other laboratory magnetic noise sources.

A thermally insulating box was constructed from 2" thick extruded polystyrene rigid foam insulation with a double layer base and removable lid, creating a controlled temperature environment for the fluxgate sensors. Four foam tabs were glued to the floor of the box to provide a repeatable placement and alignment location for each fluxgate sensor. The analog sensor cable

was passed through a hole in the sidewall of the box and was wrapped in additional insulation to reduce heat flow. The fluxgate electronics provide an analog input for a sensor temperature measurement. A common Analog Devices LM34 temperature sensor integrated circuit, on a small separate printed circuit board, was used to measure temperature for all six experimental sensors. The LM34 was taped to the base of each sensor, adjacent to the sense coil aligned with the sinusoid magnetic test signal.

Approximately 1.1 kg of dry ice was used to cool the sensor for each trial, to ensure comparable thermal cycles. This cooled the sensor to approximately -40 °C in about four hours, when the majority of the dry ice had sublimated and the temperature stabilised. The sensor warmed back towards room temperature (~21 °C) over the following 20 hours, reaching about 15 degrees before the experimental run would be terminated and reset. The temperature of the electronics was monitored by a second temperature sensor built into the STE magnetometer electronics, and was measured to vary by less than $\pm 1$ °C during the

period of most experimental runs, with a worst case of $\pm 2$°C .

Figure 8 shows a schematic of the experimental setup. The controlled source magnetic test signal was generated by placing the sensor, in the insulating foam box, inside a Helmholtz coil consisting of two circular 66.4 cm diameter coils of ~54 turns, each with a total series resistance of 3.2 ohms. A 5k000 0.2 ppm °C$^{-1}$ resistor was placed in series with the coils. A 12 V$_{rms}$ at 1 Hz sine wave from a Stanford Research DS360 ultra low distortion function generator created a 234 nT$_{rms}$ test signal that

was used for all sensor tests.

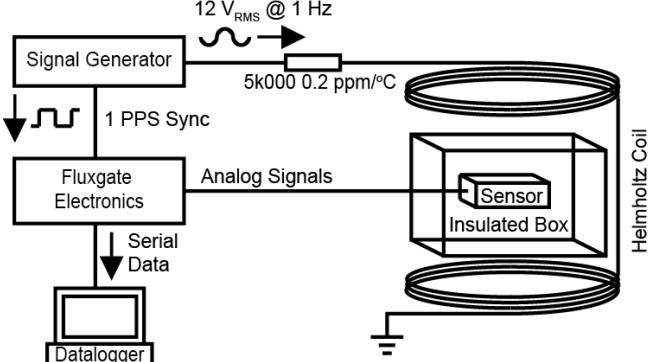

Figure 8: Experimental setup - each sensor was placed in an insulating foam box within a Helmholtz coil. The coils were driven with a sinusoidal magnetic test signal which was phase locked to the magnetometer electronics using the 1 pulse per second GPS timing input. Dry ice was used to cool the sensor and measurements were taken as the sensor warmed. The sensitivity of the sensor changes with temperature causing the measured amplitude of the constant test signal to vary with the sensor's temperature.

The small temperature coefficient of resistance (0.2 ppm °C$^{-1}$) of the resistor was intended to reduce the effect of the much larger temperature coefficient (3930 ppm °C$^{-1}$) of the copper wire in the Helmholtz coil. The stable series resistor limits the change in the applied signal due to the worst case $\pm 2°C$ room temperature variation to

$$\frac{5000\,\Omega \times 0.2\,ppm\,°C^{-1} \times \pm 2\,°C + 3.1\,\Omega \times 3930\,°C^{-1} \times \pm 2\,°C}{5000\,\Omega + 3.1\,\Omega} = 9.8\,ppm \tag{9}$$

The combined temperature coefficient of resistance of the copper coil and series resistance should contribute no more than

$$\frac{9.8\,ppm}{15\,°C - (-40\,°C)} = 0.2\,ppm\,°C^{-1} \tag{10}$$

over the 55 °C temperature range in the experiment.

Some experimental runs were contaminated due to severe local interference such as construction crews vibrating the building. However, useful data were obtained for greater than 80% of trials despite the presence of typical ambient magnetic noise in the University laboratory environment during the calibration tests.

Figure 9 shows the sensor under test in the insulating foam box. The lid, the plastic bag, and the desiccant have been removed for the photograph. Fluxgate data were gathered during the warming period of each experimental run. The temperature was cycled three times for each measurement axis, to check the consistency of the results and to estimate the error in the measurement.




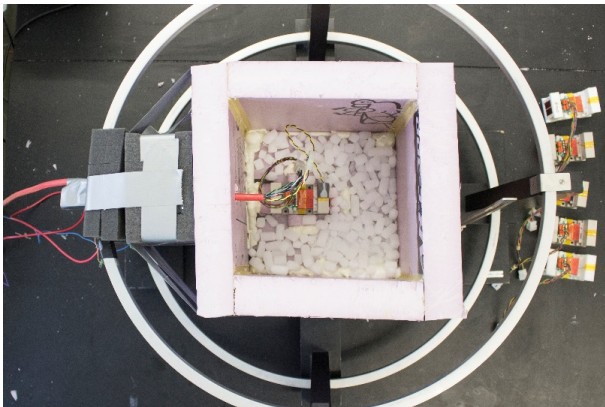

Figure 9: Sensor placed in a foam box within a Helmholtz coil. Dry ice was used to chill the sensor, and measurements were taken after the dry ice had sublimated and the sensor was slowly warming. The foam lid, plastic bag, and desiccant have been removed for the photograph.

Figure 10 shows the daily temperature variation of a sensor under test. Each morning, the dry ice was added to the foam box to start the sensor cooling. The warming rate was slow near both the minimum and room temperatures. This generated a large amount of data that dominated the fit line, compared to the middle temperature range where the rate of warming was faster. Therefore, data collection for each run was started at 0.5 °C above the minimum observed temperature. Similarly, the experiment was timed to end when the sensor was a few degrees below room temperature giving a usable warming interval

between approximately -40 °C and +15 °C of about 20 hours.

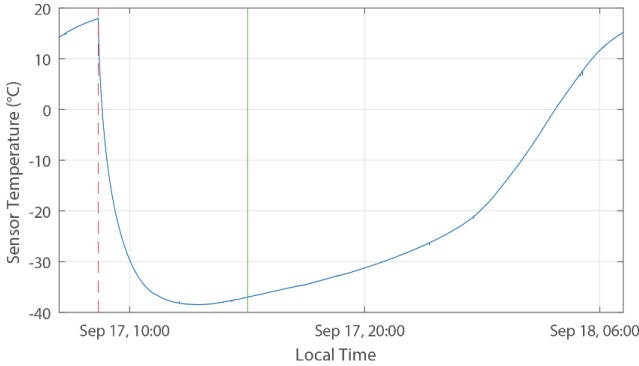

Figure 10: Temperature profile of sensor head. Dry ice was used to chill the sensor (dashed red line), and measurements were taken after the dry ice had sublimated and the sensor was slowly warming (solid green line).

The same STE magnetometer electronics were used in all the experiments. A five-meter analog cable was used to connect the
sensors to the electronics box, rather than the transmission line transformer and ~80-meter cable often used in field deployment. This configuration was selected to minimise the effect of cable length and temperature, as 80 meters of cable would not fit in


the foam box. The electronics were no longer driving a matched impedance; however, the short cable length minimised this impact.

The synchronisation output from the signal generator was driven into the pulse-per-second (PPS) timing input of the fluxgate electronics, which would normally be connected to a GPS time base. This synchronised and phase locked the 8 sps

measurements from the fluxgate to the test signal, reducing frequency beating effects.

### 3.4 Observations and Data Analysis

Figure 11 (blue) shows a Welch's averaged periodogram showing the amplitude spectrum of the magnetic noise in the laboratory. Figure 11 (red) shows the same spectra with a 1 Hz test signal applied using the Helmholtz coil. The room is magnetically noisy; however, there are no coherent sources near 1 Hz, giving a signal to noise ratio of 68 dB in the 1 Hz

frequency bin. The controlled source at 1 Hz and careful spectral analysis should therefore allow relatively noise free measurement despite the general background noise of the laboratory. The large amplitude harmonics of the controlled source was not expected and remains unexplained.

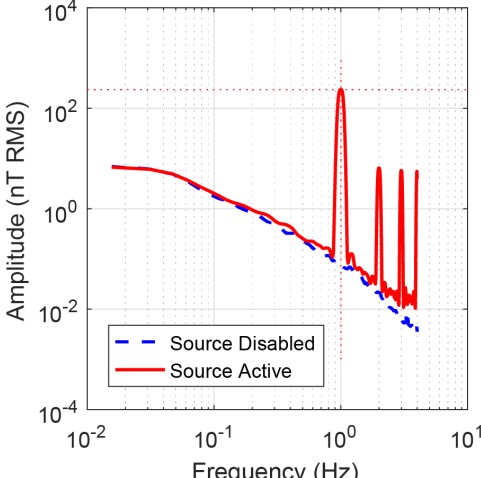

Figure 11: Amplitude spectra showing representative noise environment of the laboratory (blue; dashed). Equivalent spectra
with a sinusoidal test signal applied using the Helmholtz coil (red; solid).

The STE magnetometer generates 8 sps measurements of the magnetic field. This was analysed by subdividing into a series of non-overlapping blocks of 16384 samples (34 minutes). The choice of block length is a significant trade-off in the data analysis. Longer blocks sample more periods of the 1 Hz test signal, allowing a more precise estimation. However, a longer block length increases the chance that a transient noise event from the laboratory, with power at 1 Hz, will occur and contaminate the block.

Increasing the number of points in the Fast Fourier Transform (FFT) decreases the likelihood that environmental noise sources will fall into the same spectral bin as the test signal, but decreases averaging for a fixed block size. The measured amplitude





of the 1 Hz test signal for the block was calculated using Welch's method of overlapping periodograms (Matlab pwelch), a 512 bin FFT, an overlap of 50%, and an HFT248D flat-top window function (Heinzel et al., 2002). The data shown in Figure 11 were obtained using these parameters. The apparent magnitude of the test signal was found to be robust over block lengths between 10 and 60 minutes and FFT lengths between 128 and 1024 points suggesting the selected FFT parameters are a

reasonable compromise.

For each block of data, the sensitivity of the instrument was determined by the apparent amplitude of the test signal as measured by the FFT bin corresponding to 1 Hz in the averaged periodogram. The sensor and room temperatures were taken as the simple mean of the readings of the respective LM34 temperature sensors for the same period. Figure 12 illustrates the result for Trial 1 of the reference MACOR/MACOR sensor showing the measured amplitude of the applied magnetic test signal, the

sensor temperature, and the room temperature. Note that the Y-axis scaling of the sensor and room temperature plots are different to allow the trend in each to be observed. The sensor temperature axis spans 60 °C while the room temperature axis spans 1°C.

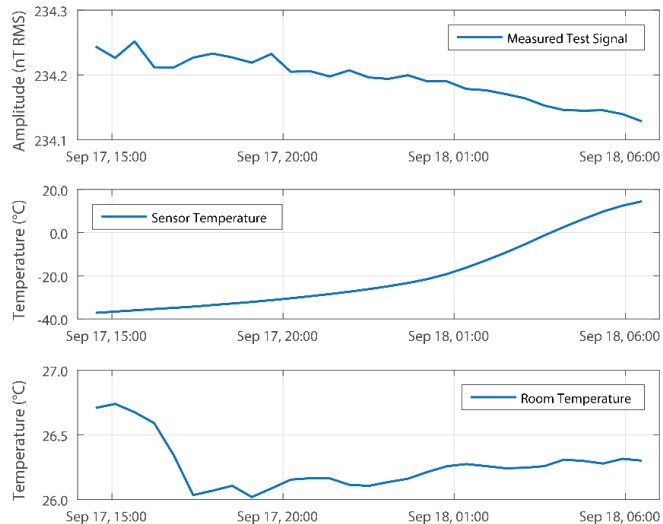

Figure 12: Trial one for MACOR/MACOR sensor. (top) Amplitude of applied sinusoidal test signal measured by the value of
the 1 Hz bin the Welch's averaged periodogram. (middle) Temperature of warming fluxgate sensor. (bottom) Room temperature.

The sensor's change in sensitivity with temperature causes the measured amplitude of the test signal to vary as the sensor warms. The coefficient of thermal gain change was determined by plotting the measured test signal amplitude against sensor temperature and fitting a linear trend. Figure 13 shows such a plot for the MACOR/MACOR sensor. Robust linear regression

(Matlab robustfit) was used for each trial to estimate a linear fit and minimize the impact of occasional outlying data. Robustfit iteratively reduces the weighting given to points away from the emerging linear trend line, allowing it to ignore the effect of

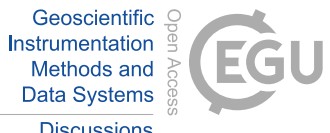



outliers. This allows the linear temperature dependence to be estimated despite outlying points contaminated with local noise from the laboratory. Each sensor was tested three times to estimate the uncertainty of the measurement. The MACOR/MACOR sensor produced linear trends with similar slopes between the three trials (-8.2 ± 0.5, -8.6 ± 0.3, -7.6 ± 0.5 ppm °C$^{-1}$). The visible Y offset between the trials are not well understood but may be related to drift in the function generator, changes in the

geometry or alignment of the Helmholtz coil, or changes in the background magnetic noise of the laboratory. However, the measured slopes agree, within experimental error, across the three trials.

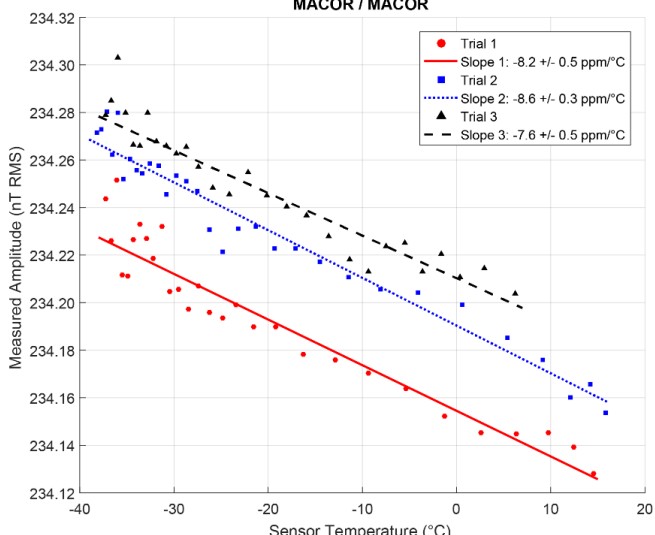

Figure 13: Results the MACOR/MACOR sensor trials 1 (red; solid; circle) 2 (blue; dotted; square) and 3 (black; dashed; triangle) for showing the measured amplitude of the test signal (scatter plot) and the linear trend (solid, dotted, and dashed
lines) determined by robust linear regression. The slope of each trial estimates the coefficient of thermal gain dependence for the sensor.

Figure 14 presents the equivalent scatter plot to Figure 13 but comparing the measured amplitude of the test signal to the room temperature when no dry ice was placed in the experiment. The scatter and shape of the room temperature data varied between trials as the building heating changed with the weather. In all cases, the room temperature was regulated to within two degrees
and no consistent trends were observed in the room temperature. This suggests that there was minimal impact from the changes in room temperature on the amplitude of the magnetic test signal created by the signal generator and Helmholtz coil.




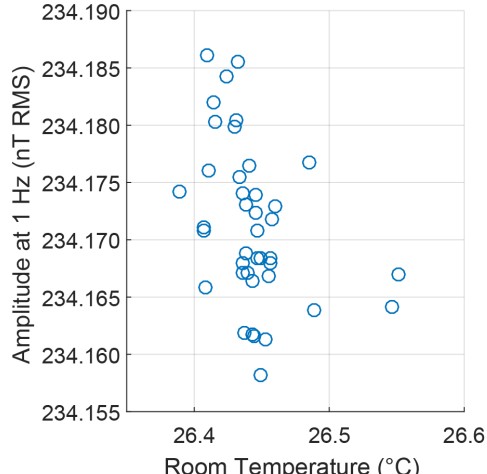

Figure 14: Measured signal amplitude plotted against room temperature. No consistent trends were observed between the different trials, suggesting that room temperature did not have a significant effect on the measured amplitude.

Part way through the experimental trials, it was discovered that the sinusoidal test signal could saturate the ADC. The test signal averages to zero at DC and therefore does not affect the instrument's decision to step the feedback offset in order to hold the instrument within range. The instrument can therefore clip when the environment's quasi-static field drifts before the instrument triggers a step in the offset voltage. Experimental trials that were clipped were discarded and repeated to obtain clean data. A smaller test signal would prevent this issue, albeit with a corresponding decrease in the signal-to-noise ratio of the measurement.

**4 Results**

Figure 15 compares the results of three trials using each of the six sensors. Y-axes are offset to account for systematic variation between the instruments but have a common span so the slopes, which give the coefficient of thermal gain dependence, can be meaningfully compared between sensors. However, the vertical span of the Y-axis is common to all plots. Therefore, the apparent slopes, which give $\alpha_g$, can be meaningfully compared. The results for each sensor are reproducible between the three runs. Conversely, the results for the six sensors vary visibly in sign, magnitude, and apparent curvature.



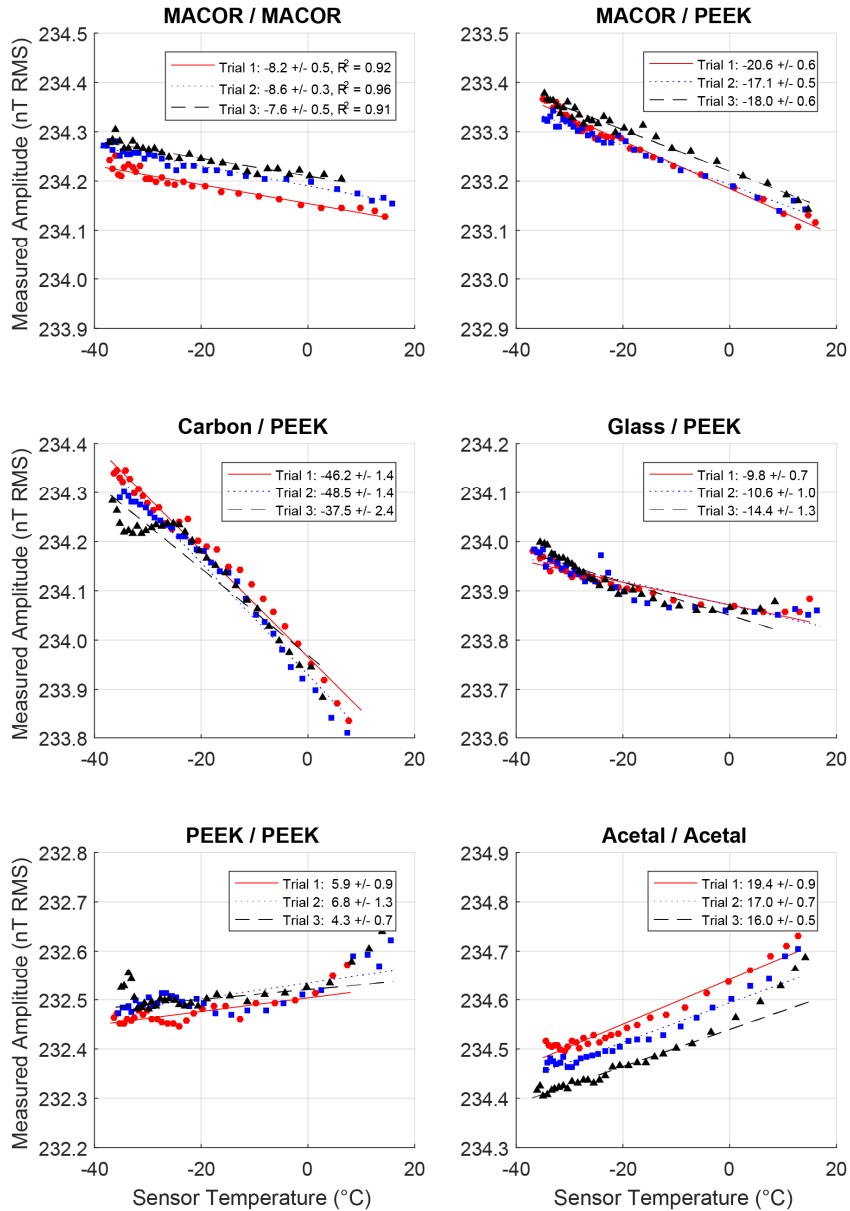

Figure 15: Measured amplitude of the constant 1 Hz test signal for three trials (red, blue, black) of all test sensors. Y-axes are offset to account for systematic variation between the instruments but have a common span so the slopes, which give the




coefficient of thermal gain dependence, can be meaningfully compared between sensors. These data are raw measurements, and have not been post-process corrected for the temperature compensation built into the electronics unit.

The measured coefficient of thermal gain dependence, $\alpha_g$, for the sensor in each trial was taken to be the slope, as determined by robust linear regression. The uncertainty in the slope was taken as plus or minus the standard error from the linear regression.

The mean of the three slopes for each sensor was plotted against the coefficient of linear thermal dependence, $\alpha_m$, of the material used in the bobbin, in order to compare the different sensors and materials (Figure 16).

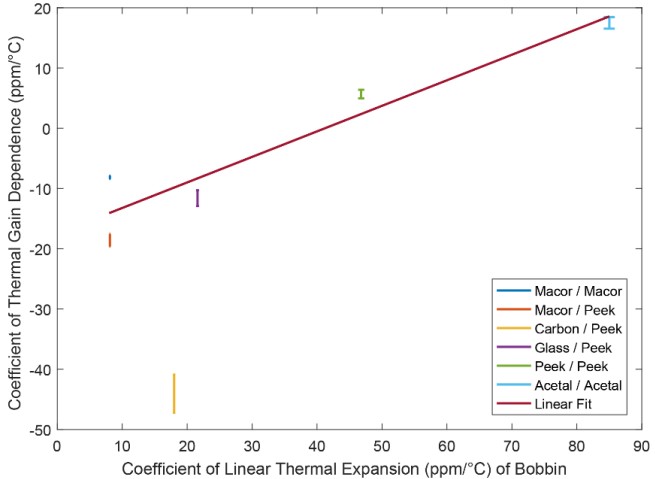

Figure 16: The measured coefficient of thermal gain dependence, $\alpha_g$, for each sensor is approximately proportional to the manufacturer specified coefficient of linear thermal expansion of the bobbin material, $\alpha_m$. Robust linear regression of the points
gives a slope of $0.42 \pm 0.09$ and an offset of $-17 \pm 4$ ppm $°C^{-1}$. The data for Carbon/PEEK is considered unreliable and was not included in the calculation of the trend line.

The Carbon/PEEK sensor had very different thermal stability than would have been expected from its specified coefficient of linear thermal expansion. As described in more detail in Appendix B, 30% carbon filled PEEK was found to be anomalously conductive, creating short-circuits within the sensor. Therefore, the results obtained for the 30% carbon filled PEEK sensor
are shown for completeness, but the Carbon/PEEK sensor has been excluded from the calculation of the trend line on Figure 16.

The uncertainty for each point in Figure 16 was calculated as plus or minus one half of the difference between the minimum and maximum values of the three trials, including the uncertainty. Robust linear regression of these points gives a slope of $0.42 \pm 0.09$ ppm $°C^{-1}$ and an offset of $-17 \pm 4$ ppm $°C^{-1}$.

The fluxgate electronics provide linear temperature compensation, so non-linear temperature effects in the sensor cannot be compensated. Quantifying the non-linear temperature dependence of the sensors is challenging due to the scatter in the data. A numerical estimate of curvature was calculated for each run, but gave high uncertainties that masked the apparent curvature





visible by eye in some runs. As an alternative, the $r^2$ linear correlation coefficient was calculated, and is provided to estimate the quality of fit for the thermal trends. However, since robust linear regression uses iterative weighting to reduce the effect of outliers, it does not minimise $r^2$ as a conventional linear fit does, so the $r^2$ coefficients are expected to be larger than would be expected from a linear fit despite the trend likely being a better fit to the data.

Table 3 summarises the results from this experiment giving: $\alpha_m$ provided by the manufacturer, which is taken to be equal to the theoretical $\alpha_g$ from (7); the measured $\alpha_g$; and an estimate of the $r^2$ linear correlation coefficient. The measured $\alpha_g$ was calculated as the mean of the fit slope from the three trials, minus the zero intercept from Figure 16 (-17 ppm °C$^{-1}$) to remove the correction applied by the electronics. The discrepancy between the theoretical and measured values of $\alpha_g$ is discussed below.

Table 3: Parameters of materials used in this study. The measured coefficient of thermal gain dependence for each sensor has been corrected by -17 ppm °C$^{-1}$ to remove the measured correction applied by the electronics. The sensor name corresponds to the material in the sensor bobbin and base, respectively.

| Sensor Name | Manufacturer's Coefficient of Linear Thermal Expansion of bobbin material / Theoretical Coefficient of Thermal Gain Dependence (ppm / °C ) | Measured Coefficient of Thermal Gain Dependence (ppm / °C ) | Coefficient of Determination $r^2$ |
|---|---|---|---|
| MACOR/MACOR | 8.1 | 9.3 ± 4 | 0.927 |
| MACOR/PEEK | 8.1 | -1.1 ± 5 | 0.979 |
| Carbon/PEEK | 18.0 | -27 ± 7 | 0.955 |
| Glass/PEEK | 21.6 | 5.9 ± 5 | 0.827 |
| PEEK/PEEK | 46.8 | 23.1 ±4 | 0.598 |
| Acetal/Acetal | 85.0 | 35.0 ± 5 | 0.950 |

## 5 Discussion

The measured coefficient of thermal gain dependency of the sensors was generally proportional to the coefficients of linear
thermal expansion of the materials used to construct the sensor's bobbins. The zero offset of -17 ± 4 ppm °C$^{-1}$ in Figure 16 implies a systematic temperature effect across all sensors that matches the 19.1 ppm °C$^{-1}$ of compensation expected from the modified transconductance amplifier within experimental error. Accounting for this compensation, the trend line would intersect zero/zero, implying that a sensor constructed from material with no deformation due to temperature would have no thermal gain sensitivity. Despite the observed difference between the MACOR/MACOR and the MACOR/PEEK sensor, this
supports the assumption that the bobbin material has the dominant effect on the $\alpha_g$ of a sensor.

The slope of the trend line in Figure 16 indicates that the coefficient of thermal gain dependence is approximately one half the coefficient of linear thermal expansion of the bobbin material, rather than being equal to it as suggested by Acuña et al. (1978).


This discrepancy seems reasonable given the simplicity of the theoretical derivation, the lack of a ferromagnetic core in the derivation, potential mechanical effects from the Inconel foil bobbin, and the succession of approximations. It would be interesting to attempt a physics driven model of the magnetic field within the sensor to see if a more sophisticated treatment would better reproduce the measured relationship between the coefficient of linear thermal expansion of the material and the

thermal gain dependence of the sensor. Such a model could also explore the relationship for other sensor geometries, such as the long circular solenoidal coils often wound on a tubular shaft containing racetrack shaped cores. Notably, the theoretical $\alpha_g$ calculated using the method of Acuña et al. (1978) agrees with the measured value of the MACOR/MACOR sensor (matching the material used in the Acuña instrument), but diverges as the $\alpha_m$ of the bobbins increase.

The significant difference between the measured $\alpha_g$ for the MACOR/MACOR sensor and the MACOR/PEEK sensor is in

some way surprising. A possible explanation might be the sensor base deforming under test as the sensor base changes temperature, thereby rotating the sense axis of the bobbin with respect to the test signal. Such an effect may be due, in part, to the mismatch in $\alpha_m$ between the material used to construct the bobbin and the base. Unfortunately, the presence of this effect cannot be tested using the current dataset. A future experiment could repeat these tests using a three component Helmholtz arrangement. Simultaneously applying test signals at three different frequencies to the three axes could test for bobbin tilt by

checking for an increase in the apparent amplitude of the test signals applied orthogonally to the nominal sense axis of the bobbin. Nevertheless, the discrepancy between the MACOR/MACOR and MACOR/PEEK sensors highlights the importance of maintaining the orthogonality of the three axes.

It is challenging to compare or interpret the linearity of the thermal dependence of the different sensors. An estimate of the $r^2$ coefficient of determination was calculated. However, it likely underestimates the quality of the fit and, does not discriminate

between scatter and nonlinear effects. By eye, it would appear that the Glass/PEEK sensor is less linear than the reference MACOR/MACOR sensor; however, the numerical uncertainties are so large that no difference can be firmly established between the linearity of the thermal dependence of the different sensors.

The measured $\alpha_g$ of the 30% glass filled PEEK bobbin was within 5 ppm °C$^{-1}$ of that of the MACOR bobbins. A sensor constructed from both 30% glass filled PEEK bobbins and a 30% glass filled base would likely have a slightly different thermal

dependence (c.f., MACOR/MACOR and MACOR/PEEK). If a modest increase in thermal dependence can be tolerated or compensated, then 30% glass filled PEEK is a good candidate for future fluxgate sensors as it is more economical, easier to machine, lighter, and more robust than MACOR.

### 6 Conclusions

1.   The coefficient of thermal gain dependence varied roughly linearly with the coefficient of linear thermal expansion of the

30       material used to support the sense and feedback windings in the STE magnetometer. However, the coefficient of thermal



gain dependence varied as approximately one half of the coefficient of linear thermal expansion of the bobbin rather than being equal as suggested by Acuña et al. (1978).

2.  The small manufacturer specified coefficient of linear thermal expansion for 30% carbon filled PEEK made it an attractive material to use. However, carbon filled peek was found to be highly conductive, and created short circuits which made the sensor unusable.

3.  30% glass filled PEEK is, on paper, modestly more sensitive to temperature than MACOR, but its robustness, cost, and ease of machining make it an attractive material for manufacturing the bobbin supporting the sensor and/or feedback windings of the fluxgate and for the base of the senor. A 30% glass filled PEEK bobbin yielded a sensor with a measured thermal coefficient of $5.9 \pm 5$ ppm °C$^{-1}$ versus $9.3 \pm 4$ ppm °C$^{-1}$ for MACOR after removing the effect of electronic temperature compensation.

4.  Highly precise (<0.1 nT) calibration measurements have been achieved in a magnetically noisy laboratory using a simple and inexpensive experimental setup comprising a constant sinusoidal magnetic test signal and temperature cycling via dry ice, and quantitative spectral analysis.

**Code availability**

Source code used to analysis and visualise the data in this manuscript is available from (Miles, 2017)

**Data availability**

Data used in this manuscript is available here: (Miles, 2017)

**Appendix A: Transconductance Amplifier Design and Sensitivity Analysis**

This document was originally prepared as an internal technical note for the University of British Columbia Department of Geophysics and Astronomy Geophysical Instrumentation Laboratory (Narod, 1982). It has been reproduced and expanded here for the public record.

**A1 Introduction**

The transconductance amplifier is one of the most critical components in a fluxgate magnetometer as it is its characteristics, combined with the thermal and electrical properties of the sensor, which determine the feedback transfer function and thus the performance of the instrument as a whole. The amplifier serves two functions. Firstly, it converts a voltage analogue of the magnetic field into a current, which in turn creates the nulling field in the sense coil. Secondly, by careful selection of the



resistors, the amplifier can be made to compensate the first order measurement distortions caused by thermal expansion of the sensor.

**A2 Circuit Analysis**

An idealized circuit for the transconductance amplifier producing an output current $I_F$ is shown in Figure A1.

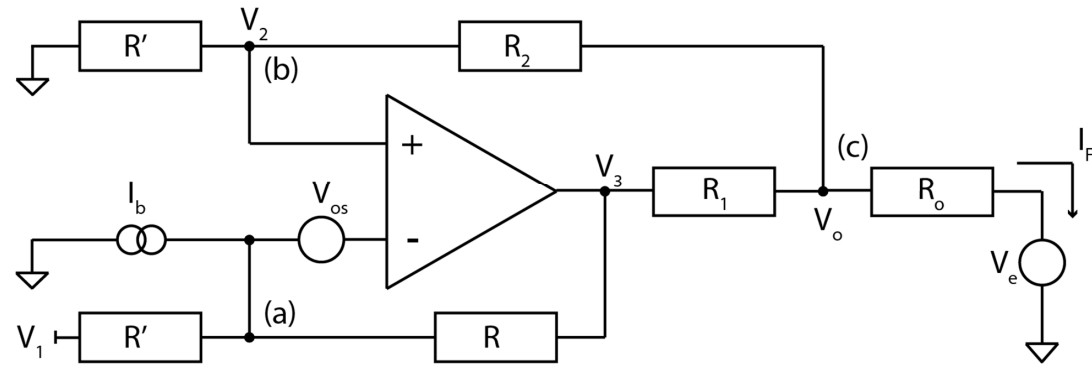

Figure A1: Idealised circuit for a transconductance amplifier.

Control voltage $V_1$ is attenuated 6 dB in the actual implementation of the transconductance amplifier by a summing node. $R_1$ is the current sense resistor. $R_o$ is the sensor load resistance. Neglecting all noise sources ($V_e$, $V_{os}$, and $I_b$), the characteristic equations from nodes (a), (b), and (c) are respectively.

$$\frac{V_1 - V_2}{R'} = \frac{V_2 - V_3}{R} \tag{A1}$$

$$\frac{V_o - V_2}{R_2} = \frac{V_2}{R'} \tag{A2}$$

$$\frac{V_3 - V_o}{R_1} = I_f + \frac{V_2}{R'} \tag{A3}$$

10    where $I_f = \frac{V_o}{R_o}$ is the feedback current flowing into the sensor. Rearranging Eqs. (A1), (A2), and (A3) gives respectively

$$R'(V_2 - V_3) = R(V_1 - V_2) \tag{A4}$$

$$R'(V_o - V_2) = R_2(V_2) \tag{A5}$$

$$R'(V_3 - V_o) = R_1 R' \frac{V_o}{R_o} + R_1 V_2 \tag{A6}$$

Combining Eqs. (A4) and (A6) to eliminate $V_3$ gives



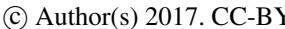


$$(R' + R - R_1)V_2 = RV_1 + R'\left(1 + \frac{R_1}{R_o}\right)V_o \tag{A7}$$

Substituting Eq. (A5) to eliminate $V_2$ gives

$$\frac{R'}{R' + R_2}(R' + R - R_1)V_o = RV_1 + R'\left(1 + \frac{R_1}{R_o}\right)V_o \tag{A8}$$

Rearranging Eq. (A8) gives

$$I_f \equiv \frac{V_o}{R_o} = \frac{-V_1\left(\frac{R}{R_1}\right)}{R_1 + R_o - R_o\left(\frac{R' + R - R_1}{R' + R_2}\right)} \tag{A9}$$

Rearranging and simplifying gives

$$I_f = \frac{-V_1\left(\frac{R}{R_1}\right)}{R_1 + R_o\left(\frac{R_1 + R_2 - R}{R' + R_2}\right)} \tag{A10}$$

The ideal transconductance amplifier has the property that $I_f$ is independent of $R_o$. From Eq. (A10), this the case when $R_1 +$

5   $R_2 - R = 0$. However, here the design goal of the transconductance amplifier is to compensate for thermal variation in the

sensor based on the resistivity coefficient of the copper sensor winding.

Defining the following parameters

$$G \equiv \frac{R}{R'} \tag{A11}$$

$$\beta \equiv \frac{R_1 + R_2 - R}{R' + R_2} \tag{A12}$$

$$R^* \equiv \left(\frac{1}{G}\right)(R_1 + \beta R_o) \tag{A13}$$

allows Eq. (A10) to be rewritten as

$$I_f = -\frac{V_1}{R^*} \tag{A14}$$





The design goal is met when $I_f$ changes with $R^*$ such that the offset field in the sensor is constant. The temperature dependence of the current output from the transconductance amplifier is

$$\frac{1}{I_f}\frac{dI_f}{dT} = \frac{-1}{R^*}\frac{dR^*}{dT} = \frac{-1}{R^*}\frac{d}{dT}\left[\frac{1}{G}(R_1 + \beta R_o)\right] \tag{A15}$$

Considering only the effect of temperature changes in the sensor then $\frac{d}{dT}R = 0, \frac{d}{dT}R_1 = 0, \frac{d}{dT}R' = 0$, and $G$ is constant with respect to temperature. Taking the temperature coefficient of resistivity of the copper in the windings to be $\alpha_c$, Eq. (8) then simplifies to

$$\frac{1}{I_f}\frac{dI_f}{dT} = \frac{-1}{R^*}\frac{1}{G}\left[\frac{d}{dT}R_1 + \beta\frac{d}{dT}R_o\right] = \frac{-1}{R^*}\frac{\alpha_c\beta R_o}{G} \tag{A16}$$

The offset field generated in the sensor is assumed to be proportional to the product of $I_f$ and the sensor turns density which is further assumed to be determined by the coefficient of linear thermal expansion, $\alpha_m$, of the bobbin which supports the sensor windings.

$$\frac{1}{I_f}\frac{dI_f}{dT} = \alpha_m \tag{A17}$$

This definition gives the following relationships

$$\beta = G\frac{-\alpha_m}{\alpha_c}\frac{R^*}{R_o} \tag{A18}$$

$$R_1 = GR^* - \beta R_o \tag{A19}$$

$$R_2 = \frac{R - R_1 + \beta R'}{1 - \beta} \tag{A20}$$

The procedure for designing the amplifier is then as follows:

1. Select $R^*$ by determining a specification

2. Choose $G, R, R'$

3. Calculate $\beta$ from Eq. (A18)

4. Calculate $R_1$ from Eq. (A19)

5. Calculate $R_2$ from Eq. (A20)



### A3 Errors on Output

Suppose an error voltage $V_e$ is placed in series with $R_o$. In the ideal transconductance amplifier, this would have no effect since it is insensitive to load variations. However, in an unbalanced amplifier, this produces a small error in the output. The analysis can be simplified by redefining the ground potential such that $V_e$ is moved from the output to both inputs giving Figure A2.

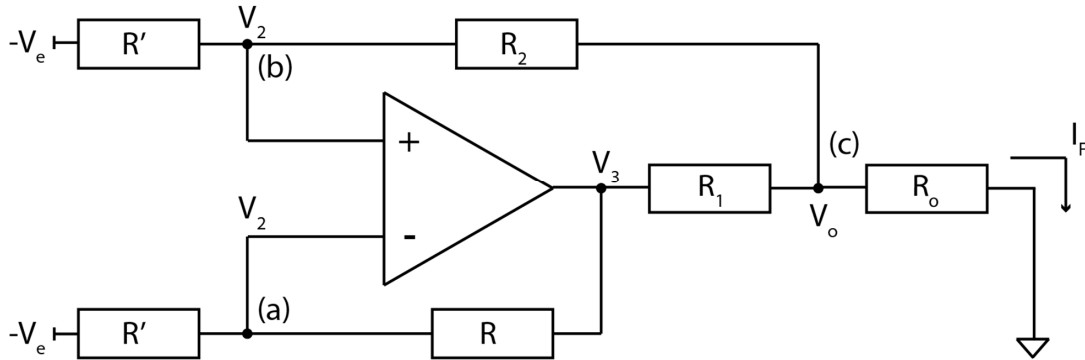

Figure A2: Error model for a transconductance amplifier.

The characteristic equations from nodes Eqs. (a), (b), and (c) respectively are then:

$$\frac{V_3 - V_2}{R} = \frac{V_2 + V_e}{R'} \tag{A21}$$

$$\frac{V_o - V_2}{R_2} = \frac{V_2 + V_e}{R'} \tag{A22}$$

$$\frac{V_3 - V_o}{R_1} = \frac{V_2 + V_e}{R'} + \frac{V_o}{R_o} \tag{A23}$$

or

$$R(V_2 + V_e) = R'(V_3 - V_2) \tag{A24}$$

$$R_2(V_2 + V_e) = R'(V_o - V_2) \tag{A25}$$

$$R'(V_3 - V_o) = R_1(V_2 + V_e) + \frac{R_1 R' V_o}{R_o} \tag{A26}$$

10    Rearranging and eliminating $V_3$ gives

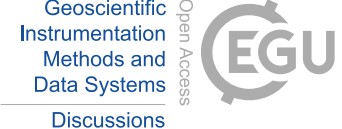

$$(R + R' - R_1)V_2 = R'\left(1 + \frac{R_1}{R_o}\right)V_o + (R_1 - R)V_e \tag{A27}$$

Eliminating $V_2$ gives

$$\frac{(R + R' - R_1)(R'V_0 - R_2V_e)}{(R_2 + R')} = R'\left(1 + \frac{R_1}{R_o}\right)V_o + (R_1 - R)V_e \tag{A28}$$

or

$$\left\{(R + R' - R_1)R' - (R_2 + R')R'\left(1 + \frac{R_1}{R_o}\right)\right\}V_o = \{(R_2 + R')(R_1 - R) + (R + R' - R_1)R_2\}V_e \tag{A29}$$

This reduces to

$$I_{fe} = \frac{V_o}{R_o} = \frac{-V_e}{R_o - \dfrac{R_1(R_2 + R')}{R - R_1 - R_2}} \tag{A30}$$

or

$$I_{fe} = \frac{-V_e}{R_o + \dfrac{R_1}{\beta}} \tag{A31}$$

5  Thus the output circuit resistance is

$$\frac{-V_e}{I_{fe}} = R_o + \frac{R_1}{\beta} = R^*\frac{G}{\beta} = -R_o\frac{\alpha_c}{\alpha_m} \tag{A32}$$

Where $Z_{out} = \frac{R_1}{\beta}$ is the amplifier output impedance.

**A4 Operational Amplifier Errors**

Operational amplifier input offset error can be analysed by considering Figure A1 with $V_1 = 0$ and an offset voltage $V_{os}$ on the inverting input. Then Eqs. (A1) and (A4) become

$$\frac{0 - (V_2 - V_{os})}{R'} = \left(\frac{V_2 - V_{os} - V_3}{R}\right) \tag{A33}$$

10  and

$$R'(V_2 - V_3 - V_{os}) = R(V_{os} - V_2) \tag{A34}$$

while Eqs. (A2), (A3), (A5), and (A6) remain constant. Defining a parameter $V^*$ such that




$$RV^* = (R + R')V_{os} \tag{A35}$$

allows Eq. (A34) to be rewritten as

$$R'(V_2 - V_3) = R(V^* - V_2) \tag{A36}$$

which is identical in form to Eq. (A4) but with $V_1$ replaced by $V^*$. Starting from Eq. (A14), and substituting $V^*$ for $V_1$ produces

$$I_{fos} = -\frac{V_{os}}{R^*}\left(1 + \frac{1}{G}\right) \tag{A37}$$

Similarly, for a bias current $I_b$ at the inverting input Eqs. (A1) and (A4) become

$$\frac{0 - V_2}{R'} = \frac{V_2 - V_3}{R} - I_B \tag{A38}$$

5 and

$$R'(V_2 - V_3) = R(-V_2) + RR'I_B \tag{A39}$$

Defining

$$V^{\#} = R'I_B \tag{A40}$$

allows Eq. (A39) to be rewritten as

$$R'(V_2 - V_3) = R(V^{\#} - V_2) \tag{A41}$$

which is identical in form to Eq. (A4) but with $V_1$ replaced by $V^{\#}$. Starting from Eq. (A14), and substituting $V^{\#}$ for $V_1$ produces

$$I_{fb} = -\frac{R'}{R^*}I_b \tag{A42}$$

10 **A5 Component Sensitivity**

Equations Eqs. (A13) and (A14) combine to give

$$I_f = \frac{-V_1 G}{R_1 + \beta R_o} \tag{A43}$$

All the component sensitivities can be derived from Eq. (A43). $\beta R_o$ is typically less than $1/100^{\text{th}}$ of $R_1$, so that absolute stability is almost chiefly dependent on the individual resistor $R_1$. The best quality resistors for the purpose are the metal foil variety with 0.5 ppm °C⁻¹ temperature coefficients and 25 ppm per year drift.



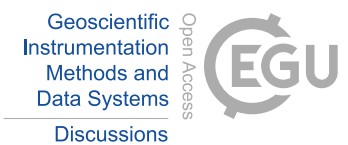

$G$ must also be stable but is principally a function of tracking stability between $R$, $R'$, and $R_2$. Typically, 25 ppm/deg absolute and 2 ppm per deg tracking stability should be adequate. Lower cost high quality metal film or thin film chip resistors should prove satisfactory.

5 The implementation of $\beta$ depends on $G$, $\alpha_c$, $\alpha_m$, $R_o$, and $R^*$ - Eq. (A18). Thus knowledge of $G$ and $R^*$ is necessary for all deterministic design methods. A heuristic design method notes that the output impedance is dependent only on $\alpha_c$, $\alpha_m$, and $R_o$ which must be known a priori. The principle function of a finite $\beta$ transconductance amplifier is to create a well defined $Z_{out}$ which can be achieved by adjusting $R_2$ to compensate for inexact implementation of $G$ and $R_1$.

To achieve first order compensation of the design range $\beta$ must be held within 1/30 of its target value. Accuracy of $\beta$ is determined mostly by the ability of $R$ and $R_2$ to track with each other Eq. (A12). By inspection of Eq. (A12), tracking 10 tolerances can be relaxed as $\beta$ is increased. Thus from Eq. (A18), $G$ should be made as large as possible. Note that Eq. (A37) implies that this also reduces $V_{os}$ sensitivity. The magnitude of G does however have a tradeoff with compliance of the amplifier.

**A6 Cabling Considerations**

For long cable runs (e.g., 100 m) cable resistance can be significant. The detrimental effect of long cables can be compensated 15 by adding a lumped platinum resistance to the sensor. This has the positive effect of lumping the temperature sensing in a very well behaved localized resistor at the sensor. However increasing the effective $R_o$ also decreases $\beta$ thus putting greater demands on the implementation of $R$ and $R_2$.

The preferred method is to have low resistance cable (<1 ohm) although a platinum sense element may also be required. Equation (A18) can then be modified to include: $\alpha_p$, the temperature coefficient of platinum; $R_p$, the resistance of the platinum 20 element; and $R_c$, the cable resistance

$$\frac{\beta}{G} = \frac{-\alpha_m}{\alpha_p} \frac{R^*}{R_o + R_p + R_c} \tag{A44}$$

where $R_p$ dominates both $R_o$ and $R_c$ and $\alpha_p$ is known to be very close to $\alpha_c$.

**Appendix B: Material Notes from Machining and Testing the Experimental Sensors**

MACOR and the three PEEK derivatives each presented unique machining challenges. The MACOR was machined using small diameter (3 mm to 8 mm) carbon steel or diamond tools, cutting fluid to control material heating, chamfered corners, 25 and shallow cuts to avoid breakage. The cutting direction was found to be very important to avoid breakage; cutting in the entry direction worked well whereas cutting on the exit direction caused surface cracking. MACOR tolerated waterjet cutting which accelerated rough cutting. However, MACOR still rapidly blunted tools, leading to slow and expensive machining.

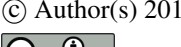



Some MACOR stock also appears to have had internal fractures which could not be spotted by eye, causing a subset of nominally identical pieces to crack under normal machining.

All three PEEK derivatives machined well using slow tool speed and liberal use of coolant to minimise heating and the resulting hardening of the machined surfaces. Virgin PEEK was particularly susceptible and the surface finish was noticeably different

from that of the as-purchased material. Virgin PEEK and 30% carbon filled PEEK both tended to deform during machining and two finishing passes were used to achieve reasonable tolerances. The 30% carbon filled PEEK also required an additional spring pass (repeating a tool path without advancing the cut) to remove protruding filaments from the machined surface. Overall, the 30% glass filled peek was the easiest to machine as it resisted surface hardening, did not accumulate surface fibres, and was less prone to deforming away from tools.

B2 Issues with 30% Carbon Filled PEEK

The manufacturer datasheets indicate that the 30% carbon filled PEEK has the best coefficient of linear thermal expansion of the PEEK derivatives (*Table 3*). However, other users had experienced these issues with the machined material causing short circuits [Werner Magnes, 2013, personal communication] and these issues were reproduced during assembly and testing. The machined 30% carbon filled PEEK bobbins were briefly flamed to remove any carbon fibers protruding from the machined

surfaces. The sense windings on the 30% carbon filled PEEK bobbins had end-to-end resistances within the expected manufacturing variability. However, all the bobbins were found to be galvanically connected to their sense winding, suggesting at least one short to the bobbin in each winding. One winding was removed and rewound but was still found to be galvanically connected to the bobbin.

During initial testing, the Carbon/PEEK sensor exhibited sudden and unpredictable changes of both sensitivity and offset

during preliminary temperature cycling. The authors speculate that residual carbon fibres on the machined bobbin surface or the crevice where machined faces meet were causing intermittent shorts as the sensor expanded, contracted, and flexed during temperature cycling. These changes stopped occurring after several temperature cycles. The authors speculate that the carbon fibres were broken by the mechanical action of the temperature cycling or the shorts had reached a steady state.

The volume resistivity of the purchased 30% carbon filled PEEK was estimated using the van der Pauw method (van der Pauw,

1958) to test the residual material from the rectangular 31x31x1.25 cm slab of material used to make the bobbin. The volume resistivity measured 0.3 $\Omega$-cm compared to the $10^{16}$ $\Omega$-cm given in the datasheet. A subsequent broader search of datasheets for comparable materials from several vendors and manufacturers found large variations in parameters including resistivity values ranging from $10^5$ to $10^{16}$ $\Omega$-cm.

Overall, the 30% Carbon filled PEEK was found to be significantly different from the material specified in the manufacturers

data sheet and appears to have caused shorting in the sensor. The authors consider it a poor candidate for future sensors and consider the data taken with the experimental Carbon/PEEK sensor to be unreliable.





**Author Contributions**

D. M. Miles led the experiment, designed and built the experimental apparatus, executed the experiment, analysed the data, and prepared the manuscript with contributions from all authors. I. R. Mann provided supervision and funding and assisted in the interpretation of the data. A. Kale helped execute this and earlier versions of the experiment and developed the software

tools to automate the data acquisition. B. B. Narod designed the STE magnetometer, helped construct the experimental sensors, and provided the acetal/acetal sensor. J. R. Bennest assembled the experimental sensors. D. Barona created the computer aided design models used to manufacture the experimental sensors, and helped interpret the data. D. K. Milling oversaw the execution of the experiment and guided the data analysis and interpretation. M. J. Unsworth provided supervision, instrumentation, and assisted in the interpretation of the data.

**Competing Interests**

The authors declare that they have no conflict of interest.

**Acknowledgements**

Work on the project was supported by the Canadian Space Agency under grant 13SUGONGEN. D. M. Miles is supported by an NSERC PGSD graduate scholarship and by funding from the Canadian Space Agency. I. R. Mann is supported by a

Discovery Grant from Canadian NSERC. The authors wish to thank A. Vinagreiro and P. Zimmerman for machining the components in the experimental sensors and C. Z. Miles for helping construct the thermal test chamber.

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
