# Peer review of "The Effect of Construction Material on the Thermal Gain Dependence of a Fluxgate Magnetometer Sensor"

_Geoscientific Instrumentation, Methods and Data Systems, 2017_

## Referee Comment (RC1) · M. Moldwin (Referee) · 16 May 2017

This paper describes a systematic test on the thermal properties of material used for the base and bobbins to hold wire windings for fluxgate magnetometers. It is inspired by a desire to find a replacement for MACOR ceramic that is very difficult to machine. MACOR was used in many early magnetometers especially those from Acuna/NASA-GSFC. The paper is a good reference for not only validating the use of PEEK for the sensor housing, but also provides in the appendix useful material that was previously found in a difficult to find Report.

(1) Title. Suggest replacing "Construction" with "sensor housing" or "winding and core support" or some other words to describe what the material is used

for "Bobbin and Winding Support"? (2) Line 10, add a comma before "but" (3) Page 2, Line 25, the Ukrainian's have looked at the temperature dependence of material (including MACRON) on fluxgate gain though they don't publish in easy to find journals. KOREPANOV, V. and MARUSENKOV, A.: Modern fluxgate magnetometers design, International Conference on Magnetism, Geomagnetism and Biomagnetism : Conference Proceedings, Sezana, 2008. p. 31-36. http://www.viviss.si/download/viviss/ZBORNIK%20MGB/Korepanov_paper_31_36.pdf (4) Page 2, Line 33. Though the sensor housing material is not explicitly named in publications, UCLA's fluxgate magnetometers have used Lexan (DSX) and have moved to PEEK. Mark B. Moldwin, "Vector Fluxgate Magnetometer (VMAG) Development for DSX," UCLA, Final Report, June 3, 2010, URL: http://www.dtic.mil/cgi-bin/GetTRDoc?Location=U2&doc=GetTRDoc.pdf&AD=ADA529004 for exactly the reasons described in this research article (cost, ease of machining, and good thermal behavior). (5) Page 8, Line 5, "Reference values..." (6) Page 11, Line 20, to be consistent with British English Spelling used throughout "Characterised" (7) Page 12, Line 12. To be consistent, use SI (or at least mm/cm) as used in other dimensions in text instead of symbol for inch. (8) Figure 11. What reasons have you eliminated for the large amplitude harmonics at 2, 3, and 4 Hz seen in the data? Is there a mismatch in the driver circuit that isn't exactly tuned to give 1 Hz signals? (9) Figure 13, Are there any other potential explanations of the Y offsets of the different trials? Any simple tests that you can do? Is the alignment of the Helmholtz coils thought to be due to a temperature effect? Do you get similar variations of off-set without the thermal test set up? (10) Page 23, Line 17, "Data are..." (11) Page 30, Line 16 "localised" (12) Page 31, Line 8, "PEEK"

---

## Referee Comment (RC2) · Anonymous Referee #2 · 11 Jul 2017

Following comments may be made. 1. There is said in the text that main influence on the transfer function has measuring winding, whereas it is known that main influence has feed-back winding. 2. The calibration using AC field is not new – it was proposed still in $\sim$ 1985 (see Yu.Afanasyev, Fluxgate sensors, 1986). 3. The tests conducted are far from exact ones: a) Temperature influence on Helmholtz coils geometry and reference signal stability are not discussed; b) It is an erroneous conclusion from the data ginen at Fig.14 about negligible influence of the room temperature on measurements precision. Even at very small change (say, 0.05 - 0.07 Đą) the output signal drift at 1 Hz was $\sim$ -100 ppm, what corresponds to -(1500...2000) ppm/C instability of test signal. At the given change of room temperature $\sim$ 2 C expected instability of

test signal can be $\sim$ -(3000...4000) ppm, what may considerably spoil measurements results. c) The signal amplitude is too small. The deviation in 100 ppm corresponds to the output signal deviation only at 234 nT/10000 $\sim$ 23 pT. 4. The possible change of the sensor orientation in thermal chamber at temperature change is not discussed. 5. There is no explanation of too high harmonic content in test signal, what may influence measurements precision. 6. The conclusion of the thermal drift value equal to about a half of thermal expansion factor of the material seems to be a partial case only for the described sensor construction. There may be several other influencing factors, would be good to discuss. The thermal drift of the compensation winding field is made only for a point in the solenoid center, what considerably differs from real sensor geometry. 7. There is a small difference of thermal factors for the sensors from macor and glass plastic, whereas the properties of these materials differ much more. Need to be explained. 8. Only one component with each material were tested what is not representative. Necessary to have statistics – as practice shows, even the properties of sensors from the same material may differ considerably. 9. The reference below cannot be reached: Miles, D. M.: Data and Source Code for: The Effect of Construction Material on the Thermal Gain Dependence of a Fluxgate Magnetometer Sensor, [online] Available from: http://dx.doi.org/10.7939/DVN/10993, 2017. We believe that these comments will be useful for authors.

---

## Author Comment (AC1) · 6 Sep 2017

We thank Dr. Moldwin for his constructive comments which we have incorporated into the manuscript. Dr. Moldwin raised several methodological questions which we address below; Dr. Moldwin's comments are in plain text, our responses in *italics* and any content added to or change in the manuscript are in *"quoted italics".*

This paper describes a systematic test on the thermal properties of material used for the base and bobbins to hold wire windings for fluxgate magnetometers. It is inspired by a desire to find a replacement for MACOR ceramic that is very difficult to machine. MACOR was used in many early magnetometers especially those from Acuna/NASA- GSFC. The paper is a good reference for not only validating the use of PEEK for the sensor housing, but also provides in the appendix useful material that was previously found in a difficult to find Report.

(1) Title. Suggest replacing "Construction" with "sensor housing" or "winding and core support" or some other words to describe what the material is used for "Bobbin and Winding Support"?

*Change made – The title now reads "The Effect of Winding and Core Support Material on the Thermal Gain Dependence of a Fluxgate Magnetometer Sensor"*

(2) Line 10, add a comma before "but"

*Change made – the text now reads "… and space physics, but are typically sensitive"*

(3) Page 2, Line 25, the Ukrainian's have looked at the temperature dependence of material (including MACRON) on fluxgate gain though they don't publish in easy to find journals. KOREPANOV, V. and MARUSENKOV, A.: Modern flux- gate magnetometers design, International Conference on Magnetism, Geomag- netism and Biomagnetism : Conference Proceedings, Sezana, 2008. p. 31-36. http://www.viviss.si/download/viviss/ZBORNIK%20MGB/Korepanov_paper_31_36.pdf

*The suggested reference (Korepanov and Marusenkov, 2008) has been added to show the historical context of the adoption of MACOR in fluxgate sensors. Korepanov and Marusenkov, (2008) specifically reference the small magnetometer in low-mass experiment (SMILE) instrument which is already referenced later in the paragraph as Forslund et al., (2008).*

*Change made – the next now reads "MACOR machinable ceramic has been used extensively and successfully in a variety of fluxgate applications often as a substitute for marble or quartz (e.g., Korepanov and Marusenkov, 2008)."*

(4) Page 2, Line 33. Though the sensor housing material is not explicitly named in publications, UCLA's fluxgate magnetometers have used Lexan (DSX) and have moved to PEEK. Mark B. Moldwin, "Vector Fluxgate Magnetometer (VMAG) Development for DSX," UCLA, Final Report, June 3, 2010, URL: http://www.dtic.mil/cgi- bin/GetTRDoc?Location=U2&doc=GetTRDoc.pdf&AD=ADA529004 for exactly the reasons described in this research article (cost, ease of machining, and good thermal behavior).

*That is a useful addition especially since, as mentioned by Dr. Moldwin, the material is not explicitly named in the publication.*

*Change made – the text now reads "… the miniaturised SMILE instrument (Forslund et al., 2007); the Vector Fluxgate Magnetometer (VMAG) for the Demonstration and Science Experiment program (Moldwin, 2010); a prototype radiation tolerant fluxgate (Miles et al., 2013) …"*

(5) Page 8, Line 5, "Reference values. . ."

*Change made – the text now reads "Reference values for general purpose acetal…"*

(6) Page 11, Line 20, to be consistent with British English Spelling used throughout "Characterised"

*Change made – the text now reads "the thermal effects being characterised."*

(7) Page 12, Line 12. To be consistent, use SI (or at least mm/cm) as used in other dimensions in text instead of symbol for inch.

*Change made – the text now reads "… was constructed from ~5 cm thick …"*

(8) Figure 11. What reasons have you eliminated for the large amplitude harmonics at 2, 3, and 4 Hz seen in the data? Is there a mismatch in the driver circuit that isn't exactly tuned to give 1 Hz signals?

*Reviewer 2 raised the same question so we have reproduced a common response here.*

*The harmonics don't appear on a direct measurement of the output of the signal generator and so are thought to be instrumental. The working hypothesis is that the test signal, added to the ambient magnetic field, occasionally passes the threshold at which the instrument updates the magnetic feedback in the channel under test. By design, the instrument updates the magnetic feedback at a 1 Hz cadence by updating the digital to analog converter and hence creating a step in the feedback current. The transient effect of the step in feedback current is compensated inside the instrument; however, this compensation may have been partially defeated by the removal of transmission line transformer in the experimental setup. It would be interesting to experiment with different, non-integer, test signal frequencies to try and separate the source from potential instrument ranging artifacts. Unfortunately, with this experimental setup we were restricted to a 1 Hz test signal to preserve the source synchronization and prevent frequency beating in the spectral analysis.*

*As acknowledged in the manuscript, "the large amplitude harmonics of the controlled source was not expected"; however, we argue that one of the strengths of the spectral analysis technique is that power at other frequencies should not affect the measurement. Regardless of how the power is getting into the harmonics it will be excluded by the quantitative spectral analysis as it is well separated from the source in frequency space. Further, assuming that the harmonics are an instrumental effect, such as from feedback updates, because the instrument hardware is constant throughout the experiment the comparison between the sensors should be unaffected.*

*Change made – the relevant text now reads "The large amplitude harmonics of the controlled source were not expected and remains unexplained. The authors suspect the harmonics may result from the instrument updating the digital magnetic feedback, which also occurs at a 1 Hz cadence, in response to the 1 Hz test signal aggravated by a impedance mismatch due to the removal of the transmission line transformer. However, because the instrument hardware is constant throughout the experiment the comparison between the sensors should be unaffected especially as the harmonics are well-separated form the test signal in frequency and will excluded by the spectral analysis."*

(9) Figure 13, Are there any other potential explanations of the Y offsets of the different trials? Any simple tests that you can do? Is the alignment of the Helmholtz coils thought to be due to a temperature effect? Do you get similar variations of off-set without the thermal test set up?

*We suspect that the offset between trials results from a small change in the alignment between the sensor and the Helmholtz coil which is incurred when the insulating box is opened to add dry ice. The Y offsets of 0.01 to 0.04 nTrms on a 234 nTrms signal would only require only a miniscule angular offset between runs. This agrees with the data shown in Figure 14 replotted here as a function of time rather than frequency. There is scatter comparable to all experimental runs; however, there is no long term trend suggestive of, for example, drift in the source amplitude. We have amended the text to clarify and further describe the suspected cause.*

[Figure]

*Figure 1: Data from manuscript Figure 14 plotted against time rather than temperature. Note scatter in apparent amplitude but no significant long term trend.*

*Change made – the text now reads "The visible Y offsets between the trials are not well understood but may result from a small change in the alignment between the sensor and the Helmholtz coil that is incurred when the insulating box is opened to add dry ice. In the absence of temperature change or accessing the box, the measured amplitude of the test signal (e.g., the data shown in Figure 14) has scatter in the apparent amplitude but no long term trend suggestive of effects like drift in the source amplitude which would affect the results of the test."*

(10) Page 23, Line 17, "Data are. . ."

*Change made – the text now reads "the data in this manuscript are available from …"*

(11) Page 30, Line 16 "localised"

*Change made – the text now reads "well behaved localised resistor"*

(12) Page 31, Line 8, "PEEK"

*Change made – the text now reads "Overall, the 30% glass filled PEEK was"*

*We believe we have addressed the concerns of the Reviewer and hope that Dr. Moldwin can now recommend our manuscript for publication in Geoscientific Instrumentation, Methods and Data Systems.*

---

## Author Comment (AC2) · 6 Sep 2017

We thank the Reviewer for their constructive comments which we have incorporated into the manuscript.   The Reviewer raised several methodological questions which we address below; the Reviewers comments are in plain text, our responses in *italics* and any content added to or changed in the manuscript are in *"quoted italics".*

Following comments may be made.

1. There is said in the text that main influence on the transfer function has measuring winding, whereas it is known that main influence has feed-back winding.

*The instrument used in the experiment, a Narod Geophysics Ltd. STE magnetometer, uses a single winding to sense and to provide magnetic feedback (e.g., Figure 7). We agree that the sentences in the Abstract and the Introduction might be misleading and have added clarification.*

*Change made – Page 1, Line 12 now reads "changes in the geometry of the wire coils that sense the magnetic field and/or provide magnetic feedback"*

*Change made – Page 2, Line 5 now reads "variations in the geometry of the coils of wire used to sense the magnetic field and/or provide magnetic feedback"*

2. The calibration using AC field is not new – it was proposed still in 1985 (see Yu.Afanasyev, Fluxgate sensors, 1986).

*We agree with the reviewer that the use of an AC signal for calibration is not, by itself, new. However, its application here to characterize the subtle thermal dependence of the instrument is novel and powerful as it allows the calibration to be done at low cost, with a simple experimental apparatus, and in a noisy environment. We have added clarification to the text including the suggested reference.*

*Change made – Page 12, Line 6 now reads "Here, we demonstrate a novel and low-cost method of measuring thermal gain sensitivity at the ppm $°C^{-1}$ level in an uncontrolled, magnetically noisy laboratory using a simple controlled sinusoidal source and apply the technique to characterising and comparing sensor constructed from different materials. AC field calibration has been the method of choice by two of the authors (Narod, Bennest) since 1981 using FFT spectrum analyzers, and the method has also been developed independently by others (e.g., Afanas'ev, 1986)."*

3. The tests conducted are far from exact ones:

a) Temperature influence on Helmholtz coils geometry and reference signal stability are not discussed;

*The reviewer is correct, especially in light of further analysis performed in responsive to (3b). Analysis and discussion have been added as part of the response to (3b) below.*

*Change made – see (3b) below.*

b) It is an erroneous conclusion from the data ginen at Fig.14 about negligible influence of the room temperature on measurements precision. Even at very small change (say, 0.05 - 0.07 Ða̧) the output signal drift at 1 Hz was ~ -100 ppm, what corresponds to -(1500...2000) ppm/C instability of test signal. At the given change of room temperature ~ 2 C expected instability of test signal can be -(3000...4000) ppm, what may considerably spoil measurements results.

*The reviewer is correct and further analysis of the data used in Figure 14 shows a small, but non-zero, negative correlation between room temperature and the measured value of the nominally constant test signal. We have expanded Figure 14 and acknowledged this limitation of the current test fixture in the text. We thank the reviewer for pointing out this import effect in our data and limitation of our experimental apparatus.*

*Changes made – the relevant text and figure are now:*

*"Figure 14 (left) presents the equivalent scatter plot to Figure 13 but compares the measured amplitude of the test signal to the room temperature when no dry ice was placed in the experiment. The scatter and shape of the room temperature data varied between trials as the building heating changed with the weather. In all cases, the room temperature was regulated to within two degrees. Figure 14 (right) shows the same data but compares the progression of measured signal amplitude and room temperature over time. The measured signal amplitude was found to have a weak, but non-zero, anti-correlation with room temperature (R=-0.38, $R^2$ = 0.15, p = 0.013). It seems likely that this is due to slight changes in the geometry of the Helmholtz coil with temperature, potentially as the aluminum frames supporting the Helmholtz coils expanded and contracted with temperature slightly changing the effective area of the coil. The period of the room's temperature fluctuations (~2 hours) is significantly shorter than the temperature swing used in the experiment (~12 hours) so the effect on the fitted linear trend in Figure 13 and Figure 15 should be modest. However, this dependency on room temperature is not ideal and should be further mitigated in future versions of the experiment.*

[Figure]

*Figure 14: (Left) Measured signal amplitude plotted against room temperature. (Right) Measured signal amplitude and room temperature over time. The measured signal amplitude was found to have a weak, but non-zero, anti-correlation with room temperature (R=-0.38, $R^2$ = 0.15, p = 0.013).*

*"*

c) The signal amplitude is too small. The deviation in 100 ppm corresponds to the output signal deviation only at 234 nT/10000 23 pT.

*There is an instrumental limit to the maximum size of the test signal which is imposed by the offsetting design of the instrument where variable offset current is used to null the magnetic field within the sensor and extend the range of the instrument. The forward path of the instrument has a dynamic range which corresponds to ~800 nTpp. The 234 nTrms test signal (661 nTpp) is about as large as practical without risking saturating the instrument's forward path. The magnetometer only updates its magnetic feedback once a second so the 1 Hz test signal (1 Hz is currently required for synchronization between the source and the magnetometer to prevent frequency beating) cannot be tracked using the magnetic feedback and must be accommodated by the instruments forward path.*

*However, as shown in Figures 12, 13, and 15, the technique of using a phase locked controlled source and quantitative spectral analysis using Welch's method can repeatedly resolve a trend with a total deviation of less than 200 pT corresponding to ppm/°C resolution over the ~60° temperature range of the experiment. We have added explanatory text to the Experimental Setup.*

*Change made – inserted text reads "There was an instrumental limit to the maximum possible amplitude of the test signal imposed by the offsetting design of the instrument. The instrument's forward path has a dynamic range which corresponds to ~800 nTpp. The 234 nTrms test signal (661 nTpp) is as large as practical without risking saturating the instrument's forward path. The magnetometer only updates its*

*magnetic feedback once a second so the 1 Hz test signal (1 Hz is required for synchronization between the source and the magnetometer to prevent frequency beating) cannot be tracked using magnetic feedback and must be accommodated by the instrument's forward path."*

4. The possible change of the sensor orientation in thermal chamber at temperature change is not discussed.

*We have added discussion of sensor orientation as part of our response to the Reviewer's Point 7 below.*

5. There is no explanation of too high harmonic content in test signal, what may influence measurements precision.

*We agree that the high harmonic content is unexpected and undesirable. Dr. Moldwin raised the same question so we have reproduced a common response here.*

*The harmonics don't appear on a direct measurement of the output of the signal generator and so are thought to be instrumental. The working hypothesis is that the test signal, added to the ambient magnetic field, occasionally passes the threshold at which the instrument updates the magnetic feedback in the channel under test. By design, the instrument updates the magnetic feedback at a 1 Hz cadence by updating the digital to analog converter and hence creating a step in the feedback current. The transient effect of the step in feedback current is compensated inside the instrument; however, this compensation may have been partially defeated by the removal of transmission line transformer in the experimental setup. It would be interesting to experiment with different, non-integer, test signal frequencies to try and separate the source from potential instrument ranging artifacts. Unfortunately, with this experimental setup we were restricted to a 1 Hz test signal to preserve the source synchronization and prevent frequency beating in the spectral analysis.*

*As acknowledged in the manuscript, "the large amplitude harmonics of the controlled source were not expected"; however, we argue that one of the strengths of the spectral analysis technique is that it also should not affect the measurement. Regardless of how the power is getting into the harmonics it will be excluded by the quantitative spectral analysis as well separated from the source in frequency space. Further, assuming that the harmonics are an instrumental effect, such as from feedback updates, because the instrument hardware is constant throughout the experiment the comparison between the sensors should be unaffected.*

*Change made – the relevant text now reads "The large amplitude harmonics of the controlled source was not expected and remains unexplained. The authors suspect the harmonics may result from the instrument updating the digital magnetic feedback, which also occurs at a 1 Hz cadence, in response to the 1 Hz test signal aggravated by a impedance mismatch due to the removal of the transmission line transformer. However, because the instrument hardware is constant throughout the experiment the comparison between the sensors should be unaffected especially as the harmonics are well-separated from the test signal in frequency and will be excluded by the spectral analysis."*

6. The conclusion of the thermal drift value equal to about a half of thermal expansion factor of the material seems to be a partial case only for the described sensor construction. There may be several other influencing factors, would be good to discuss. The thermal drift of the compensation winding field is made only for a point in the solenoid center, what considerably differs from real sensor geometry.

*We agree with the reviewer that one-half scaling of relationship between the coefficient of linear thermal expansion of the material and the thermal stability of the sensor is likely tied to the sensor geometry and will likely not generalize directly; we have added more explicit acknowledgment of this limitation to the manuscript.*

*Change made – the relevant text in Section 5 now reads: "The slope of the trend line in Figure 16 indicates that the coefficient of thermal gain dependence is approximately one half the coefficient of linear thermal expansion of the bobbin material, rather than being equal to it as suggested by Acuña et al. (1978) for this particular bobbin geometry."*

*We also agree with the reviewer that the discussion of other potential influencing factors is important, and we included this in the original paper on page 22, lines 1-2.*

*No change made.*

*We further agree with the reviewer that our model for the effect of temperature on the compensation winding is quite primitive as it only considers a single point. We would welcome a more sophisticated model of these effects (cf. page 22, lines 2-6) but consider it beyond the scope of this, experiment focused, manuscript. We have added acknowledgments of the simplicity of our model in two additional places in the manuscript.*

*Change made – the relevant text in Section 5 now reads: "It would be interesting to attempt a physics driven model of the magnetic field within the sensor, rather than considering only a single point for simplicity, to see if a more sophisticated treatment would better reproduce the measured relationship between the coefficient of linear thermal expansion of the material and the thermal gain dependence of the sensor. Such a model could also explore the relationship for other sensor geometries, such as the long circular solenoidal coils often wound on a tubular shaft containing racetrack shaped cores."*

*Change made – the relevant text in Section 6 Conclusions now reads "… However, the coefficient of thermal gain dependence varied as approximately one half of the coefficient of linear thermal expansion of the bobbin rather than being equal as suggested by Acuña et al. (1978) for this particular bobbin geometry."*

7. There is a small difference of thermal factors for the sensors from macor and glass plastic, whereas the properties of these materials differ much more. Need to be explained.

*We agree with the reviewer that the disproportionate difference in thermal stability of the MACOR/PEEK sensor compared the Glass/PEEK sensor suggests something interesting about the thermal effects in the sensor. We suspect it is related to the difference in thermal stability between the MACOR/MACOR sensor*

*and the MACOR/PEEK sensor despite them having bobbins manufactured from the same material. We hypothesize that the mismatch in thermal expansion between the bobbin and the base of the sensor may be causing the outer two bobbins to skew away from orthogonal as the temperature changes. However, we cannot show this definitively using the current dataset.*

*Change made – we have introduced the following text and figure:*

*"Future work should also determine if filling the void above the center bobbin with the sensor base material increases the temperature stability by improving the symmetry of the thermal expansion. This could potentially explain the discrepancy in the measured coefficient of thermal gain dependence between the MACOR/MACOR sensor and the MACOR/PEEK sensor, which have identical bobbin materials, and between the MACOR/PEEK sensor and the Glass/PEEK sensor whose bobbins are constructed from material with very similar coefficients of linear thermal expansion. The mismatch in thermal expansion between the materials in the base and the bobbin, particularly the center bobbin, may act as a wedge (Figure 17, solid arrow) which combined with the asymmetry in the shape of the base might push the axes away from orthogonal as the temperature changes (Figure 17, dashed arrow).*

[Figure]

*Figure 17: MACOR/PEEK sensor showing the asymmetry in PEEK material above and below the center MACOR bobbin. Mismatch in thermal expansion between the center bobbin (solid arrow) and the base could potentially force the two outer bobbins to skew away from orthogonal (dashed arrow) as the sensor changes temperature.*

*"*

8. Only one component with each material were tested what is not representative. Necessary to have statistics – as practice shows, even the properties of sensors from the same material may differ considerably.

*We agree that it would be desirable to test all three components of each sensor. However, in the current apparatus, where each test takes one work day, that would have required at least another 36 days of continuous testing which was beyond the amount of time available with the hardware. We are working on a generalized version of this test using a three axis Helmholtz coil where each test axis is driven at different and non-integer AC frequency. We believe this will allow us to test all three axes at once while, simultaneously, making it easier to test for skewing of the axes or rotation of the sensor. However, this is a significantly more sophisticated apparatus and is beyond the scope of this manuscript.*

*Change made – we have added to following text to the discussion "A disadvantage of the current experimental apparatus is that only one of the three sensor axes can be tested at a time which is*

*aggravated by need for slow, day long, temperature cycles. Characterising the other two axes on each sensor would have required at least another 36 days of continuous testing. However, the technique of using a controlled AC source and quantitative spectral analysis should generalise to a more sophisticated test apparatus using three nested orthogonal coil systems to apply separate, non-integer, AC test signals. In theory, this should allow for all three axes to be tested during one temperature cycle. Simultaneously, this should make it easier to test for skewing of the axes or rotation of the sensor by providing three, nominally independent, measures. "*

9. The reference below cannot be reached: Miles, D. M.: Data and Source Code for: The Effect of Construction Material on the Thermal Gain Dependence of a Fluxgate Magnetometer Sensor, [online] Available from: http://dx.doi.org/10.7939/DVN/10993, 2017.

*We apologize. Perhaps this was an intermittent fault with the repository. We have checked this from multiple places and continents and as of 2017-08-24 the data and source code repository link appears to be active:*

*http://dx.doi.org/10.7939/DVN/10993*

*The doi.org link should redirect to the institutional repository address here, and we include a screenshot obtained of the relevant webpage, which we hope that the Reviewer can now access:*

*https://dataverse.library.ualberta.ca/dataset.xhtml?persistentId=doi:10.7939/DVN/10993*

[Figure]

*We believe we have addressed the concerns of the Reviewer and hope that the Reviewer can now recommend our manuscript for publication in Geoscientific Instrumentation, Methods and Data Systems.*